# Intracellular trafficking of Notch orchestrates temporal dynamics of Notch activity in the fly brain

Miaoxing Wang [1,6], Xujun Han [1,6], Chuyan Liu [2], Rie Takayama[1], Tetsuo Yasugi[1], Shin-Ichiro Ei[3], Masaharu Nagayama [4], Yoshitaro Tanaka[5] & Makoto Sato [1,2 ✉]

While Delta non-autonomously activates Notch in neighboring cells, it autonomously inactivates Notch through *cis*-inhibition, the molecular mechanism and biological roles of which remain elusive. The wave of differentiation in the *Drosophila* brain, the 'proneural wave', is an excellent model for studying Notch signaling in vivo. Here, we show that strong nonlinearity in *cis*-inhibition reproduces the second peak of Notch activity behind the proneural wave in silico. Based on this, we demonstrate that Delta expression induces a quick degradation of Notch in late endosomes and the formation of the twin peaks of Notch activity in vivo. Indeed, the amount of Notch is upregulated and the twin peaks are fused forming a single peak when the function of Delta or late endosomes is compromised. Additionally, we show that the second Notch peak behind the wavefront controls neurogenesis. Thus, intracellular trafficking of Notch orchestrates the temporal dynamics of Notch activity and the temporal patterning of neurogenesis.

[1] Mathematical Neuroscience Unit, Institute for Frontier Science Initiative, Kanazawa University, Kanazawa, Ishikawa, Japan. [2] Laboratory of Developmental Neurobiology, Graduate School of Medical Sciences, Kanazawa University, Kanazawa, Ishikawa, Japan. [3] Department of Mathematics, Faculty of Science, Hokkaido University, Sapporo, Hokkaido, Japan. [4] Research Institute for Electronic Science, Research Center of Mathematics for Social Creativity, Hokkaido University, Sapporo, Hokkaido, Japan. [5] Department of Complex and Intelligent Systems, School of Systems Information Science, Future University Hakodate, Hakodate, Hokkaido, Japan. [6] These authors contributed equally: Miaoxing Wang, Xujun Han. ✉email: makotos@staff.kanazawa-u.ac.jp

Notch (N) signaling plays diverse roles in many biological processes[1]. N-mediated lateral inhibition is reiteratively used to select a small number of differentiated cells from a large number of undifferentiated cells in a spatially and temporally regulated manner[2]. We previously demonstrated that N-mediated lateral inhibition regulates the speed of proneural wave progression when combined with epidermal growth factor (EGF)-mediated reaction diffusion[3,4].

A membrane-bound ligand, Delta (Dl), plays major roles in N-mediated lateral inhibition[5]. It non-autonomously activates N in adjacent cells through a process "*trans*-activation." Upon binding with Dl, the intracellular domain of N is cleaved to produce the N intracellular domain (N$^{ICD}$), which forms a complex with a DNA-binding transcription regulator, Suppressor of Hairless (Su (H)), and regulates target gene transcription[6–8]. On the other hand, N is autonomously inactivated by Dl expressed in the same cell through a process "*cis*-inhibition," whose molecular mechanism and biological significance remain largely elusive[9–12].

The direct interaction between Dl and N seems to trigger *cis*-inhibition by inhibiting N prior to or following its transport to the plasma membrane[13]. There are two possible mechanisms of *cis*-inhibition. First, the *cis*-interaction of the ligand and receptor may shut off the transport of N from the endoplasmic reticulum (ER) to the plasma membrane[14]. Second, the *cis*-interaction may trigger the catalytic process that results in N degradation. For example, the Dl–N complex may be internalized from the plasma membrane to cause N degradation. Protein degradation in late endosomes has been shown to play important roles in activating and inactivating N signaling during *trans*-activation[15–24]. However, the potential roles of intracellular trafficking of Dl and N in *cis*-inhibition remain largely unknown.

On the surface of the developing fly brain, the wave of differentiation, "proneural wave," propagates along the two-dimensional sheet of neuroepithelial cells (NEs), which sequentially differentiate into neuroblasts (NBs), the neural stem-like cells. In the previous study, we formulated a mathematical model of the proneural wave, which includes N activity ($N$), Dl expression ($D$), EGF signal activity ($E$), and the state of NB differentiation ($A$). $A$ is related to the expression levels of Achaete-Scute Complex proteins (AS-C). The model successfully reproduces the complex behaviors of the proneural wave in various genetic backgrounds[25,26]. N is activated along the wavefront, forming an activity peak that negatively regulates the wave propagation[27,28]. However, N is activated again behind the proneural wave, showing twin peaks of N activity in vivo[3,29] (Fig. 1a–d). If the location of Dl-expressing cells does not change, the combination of *trans*-activation and *cis*-inhibition would robustly form the twin activity peaks of Notch[11]. However, Dl expression propagates as the proneural wave progresses. The mechanism that forms the twin peaks of N activity and the biological significance of the second N activity peak have not been addressed thus far.

Behind the proneural wave, NBs start producing diverse types of neurons. The production of neural diversity is controlled by the transition of temporal transcription factors sequentially expressed in NBs. Homothorax (Hth), Klumpfuss (Klu), Eyeless (Ey), Sloppy paired (Slp), Dichaete (D), and Tailless (Tll) are known to act as the temporal transcription factors in the developing medulla (Fig. 1b)[30,31]. While Hth expression is already upregulated in NEs prior to NB differentiation, the expression of the other temporal factors is upregulated behind the proneural wave. Thus, the proneural wave could be the initial trigger of the temporal transcription factor cascade following Klu. Since the second N peak is found in NBs behind the wavefront, N signaling could trigger the transition of temporal transcription factor expression.

In this study, we reproduce the twin peaks of N activity by modifying the previous mathematical model and demonstrate that a strong nonlinearity in *cis*-inhibition robustly reproduces the twin peaks. As a potential candidate mechanism of the nonlinear behavior of *cis*-inhibition, we assume that Dl may transport N to late endosomes, in which Rab7 and the ESCRT (endosomal sorting complexes required for transport) complex quickly degrade N, which results in the inactivation of N between the twin peaks. Indeed, partial knockdown of *Dl* or inactivation of ESCRT complex causes upregulation of N activity and fusion of the twin peaks, forming a single peak of N activity. These results support the idea that intracellular trafficking of N triggers *cis*-inhibition, which changes the dynamics of N activity. We further explore the biological significance of the twin peaks of N activity. Interestingly, the second N activity peak coincides with Klu expression in the medulla NBs. We demonstrate that Klu expression depends on Dl expression along the proneural wave-front, and the formation of the single N peak by inhibiting ESCRT function results in the abnormal temporal patterning of NBs and neurons. Thus, we demonstrate the molecular mechanism of *cis*-inhibition that orchestrates the temporal dynamics of N activity and the temporal patterning of neurogenesis.

## Results

**Nonlinear *cis*-inhibition establishes the twin peaks**. In our previous mathematical model, the *cis*-inhibition term was proportional to Dl expression[3]. When the magnitude of *trans*-activation and *cis*-inhibition is roughly equivalent, N is activated only once around the wavefront. However, N is activated again behind the wavefront, showing the twin activity peaks (Fig. 1a–d)[3,29]. We used two independent N activity markers. E(spl)mγ-GFP (mγGFP) shows nuclear signals that are more prominent in the first peak than in the second peak (Fig. 1d)[32]. In contrast, NRE-dVenus shows cytoplasmic signals that are more prominent in the second peak than in the other (Fig. 1c)[33].

If Dl is continuously expressed in the same cells, the combination of *trans*-activation and *cis*-inhibition would robustly form the twin peaks of Notch activity[11]. However, Dl-expressing cells change as the proneural wave progresses. To generate a robust gap between the twin peaks, N must be quickly inactivated just after the first activity peak. Since Dl is specifically expressed at the wavefront (Fig. 1b), Dl-mediated *cis*-inhibition could be the cause of the inactivation following the first peak and the formation of the twin peaks of N activity.

We asked if the twin peaks could be reproduced by modifying *cis*-inhibition in the mathematical model of the proneural wave. It was demonstrated that the kinetics of *cis*-inhibition are very fast compared with the gradual effect of *trans*-activation from a series of in vitro experiments[11]. Thus, we incorporated a step function term for *cis*-inhibition, with which *cis*-inhibition does not occur when Dl expression is below the threshold ($e_c$), but quickly inactivates N when Dl expression exceeds the threshold (Fig. 1f). The twin peaks pattern of N activity was reproduced without significantly compromising the magnitude of N activity for a wide variety of parameter settings (Fig. 1e, f and Supplementary Figure 1).

However, the step function is very artificial and nonbiological. According to the previous literature[11], we incorporated the Hill functions for *trans*-activation and *cis*-inhibition. The Hill function is commonly used to model a biochemical reaction, in which the Hill's coefficient ($n_t$ and $n_c$) and activation coefficient ($k_t$ and $k_c$; Fig. 1g) specifies the kinetics of the reaction. We systematically modified the Hill's and activation coefficients of *cis*-inhibition to modify its response speed (Fig. 1g

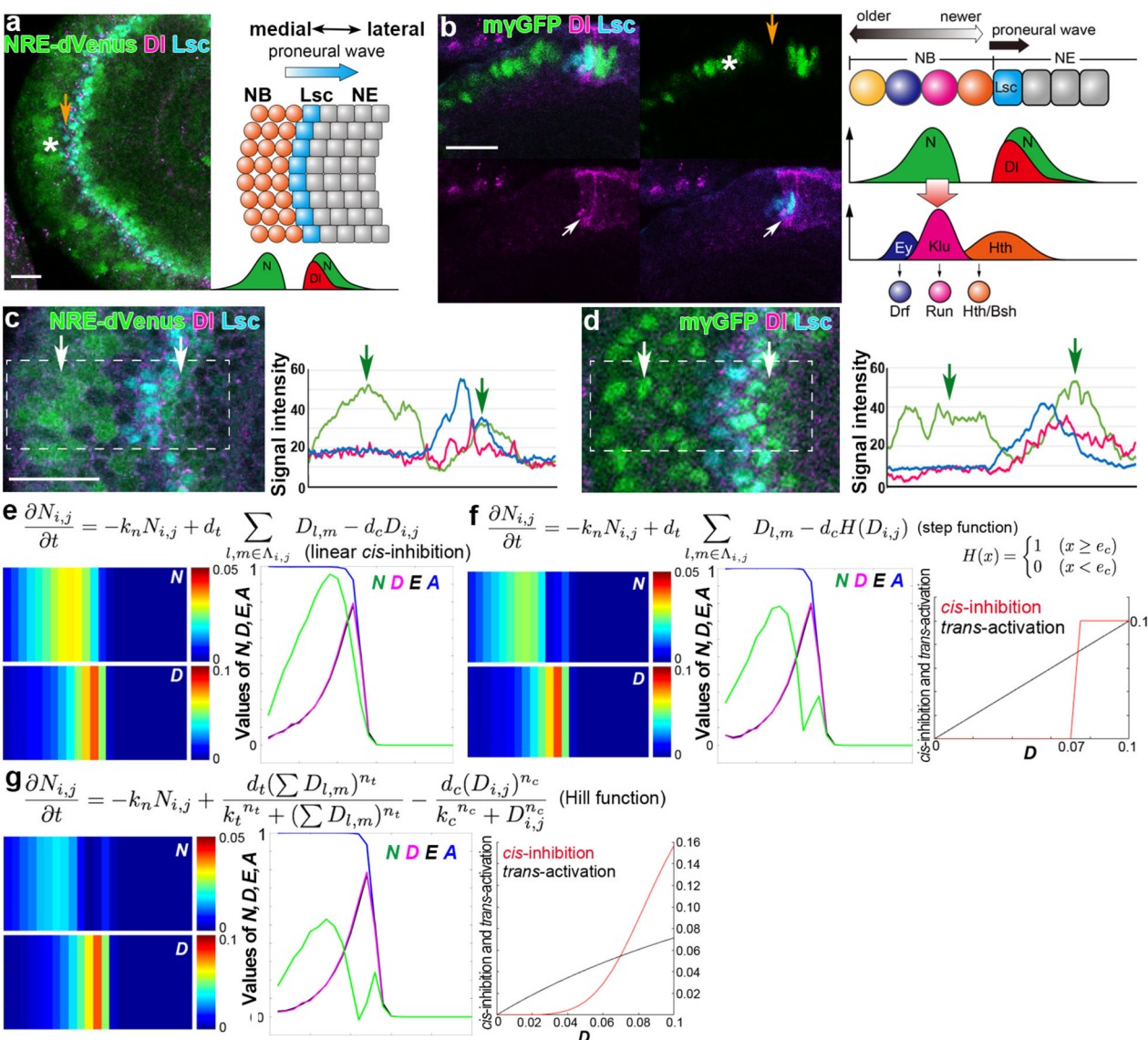

**Fig. 1 Nonlinear *cis*-inhibition establishes the twin peaks of Notch activity. a** Optic lobe in a lateral view showing the twin peaks of Notch activity (NRE-dVenus, green), wavefront (Lsc, blue), and Dl (magenta) in the left panel. The proneural wave progresses in a medial-to-lateral orientation (left to right). Asterisk indicates the second peak of N activity in NBs. Orange arrow indicates the gap between the twin peaks. The right panel shows that Lsc expression at the wavefront triggers the differentiation from NE to NB and overlaps with Dl expression and the first peak of N activity. **b** A section showing the twin peaks of Notch activity (mγGFP, green), wavefront (Lsc, blue), and Dl (magenta). White arrows show Dl expression at the wavefront. Orange arrow indicates the gap between the twin peaks of N activity. Schematic shows that NBs sequentially express temporal transcription factors, Hth (orange), Klu (magenta), and Ey (blue), which control the production of neurons expressing Hth/Bsh, Run, and Drf, respectively. **c, d** N activity is visualized by NRE-dVenus (**c**) and mγGFP (**d**) in green and compared with the distributions of Dl (magenta) and Lsc (blue). Signals in the dotted box areas are plotted in the right panels. Arrows indicate the twin peaks of N activity. Scale bars indicate 20 μm (**a–d**). **e–g** Results of numerical simulations: left panels show the two-dimensional patterns of N and D. Middle panels show the one-dimensional patterns of N (green), D (magenta), E (black), and A (blue; y = 13). The values of N, D, and E are 30, 10, and 10 times multiplied, respectively. Right panels in (**f, g**) compare the profiles of *trans*-activation (black) and *cis*-inhibition (red) in response to D. The values of *trans*-activation are four times multiplied because one cell could receive *trans*-activation from as many as four adjacent cells. **e** Single peak of N activity in the previous mathematical model in which the effect of *cis*-inhibition is linear. **f, g** Twin peaks of N activity are reproduced by introducing step function (**f**) or Hill function (**g**) to *cis*-inhibition.

and Supplementary Figure 2). The twin peaks pattern of N activity, which is similar to the in vivo pattern (Fig. 1a, c, d), was reproduced when the kinetics of *cis*-inhibition is faster compared with that of *trans*-activation.

**Delta activates Notch activity behind the wavefront.** We asked if the mechanism demonstrated in the above in silico experiments exists in vivo. As already demonstrated, the N signal is activated in cells adjacent to a clone of cells ectopically expressing Dl, while

it is efficiently inactivated in NE cells expressing Dl (Fig. 2a)[13,29]. Since Dl-expressing cells also receive *trans*-activation from surrounding Dl-expressing cells, *cis*-inhibition appears to be dominant over *trans*-activation, which is consistent with the idea that the kinetics of *cis*-inhibition are faster compared with that of *trans*-activation.

To address this idea in a loss-of-function condition, we utilized the mathematical model. If the strong expression level of Dl at the wavefront is the trigger of *cis*-inhibition, a mild reduction in Dl

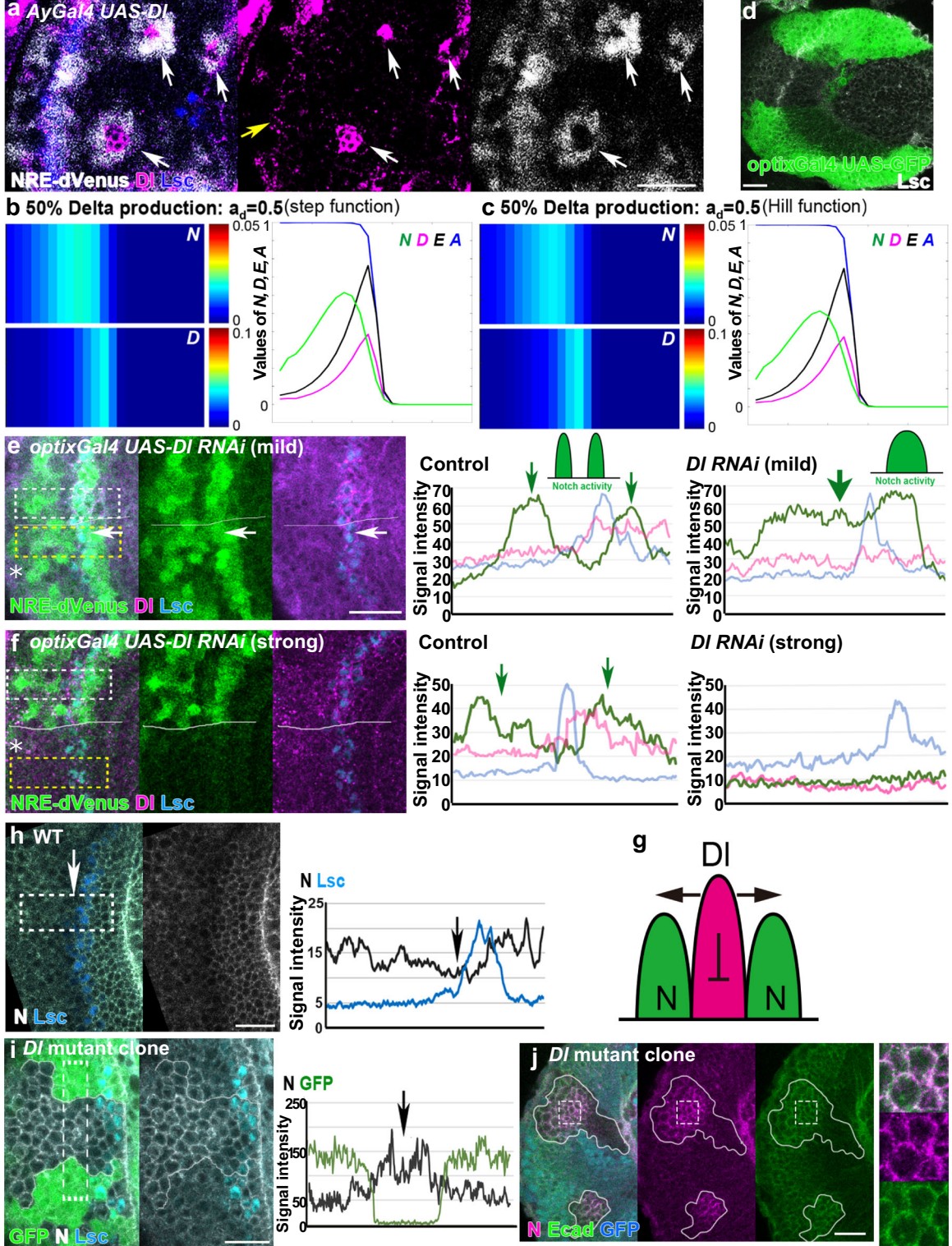

production should cause the fusion of the twin peaks, forming a single N activity peak (Fig. 2b, c). To test this idea in vivo, we made use of two different *UAS-Dl* RNA interference (RNAi) strains. When expressed in the retina under the control of *GMR-Gal4*, the mild *Dl* RNAi reduced the eye width from 0.37 to 0.35 mm, while it was further reduced to 0.31 mm by the strong *Dl* RNAi (Supplementary Figure 3).

We found that the mild *Dl* RNAi under the control of *optix-Gal4*, which is expressed in the dorsal and ventral subdomains of the optic lobe (Fig. 2d), causes the single N peak. The twin N activity peaks were fused to become a single peak (Fig. 2e and Supplementary Figure 4a). When we removed Dl expression by expressing the strong RNAi strain, both of the twin peaks of N activity were eliminated (Fig. 2f and Supplementary Figure 4b) and the same result was also observed in the *Dl* mutant clone (Supplementary Figure 4c). Note that Dl is specifically expressed at the proneural wavefront cells (Fig. 1b), suggesting that this Dl expression is responsible for the twin activation of N signaling in

**Fig. 2 Delta represses Notch activity at the wavefront. a** N activity (NRE-dVenus, white) is elevated in cells adjacent to *Dl*-expressing clones (*Ay-Gal4 UAS-Dl*, magenta), and is autonomously repressed (arrows). Yellow arrow indicates endogenous Dl expression at the wavefront (Lsc, blue). White and magenta signals are enhanced. **b**, **c** Results of numerical simulations using step function (**b**) and Hill function (**c**). Left panels show the two-dimensional patterns of N and D. Right panels are the one-dimensional plots of N (green), D (magenta), E (black), and A (blue; $y = 13$). The values of N, D, and E are 30, 10, and 10 times multiplied, respectively. Reduced production rate of D causes single peak of N activity ($a_d = 0.5$). **d** *optix-Gal4* expression in the dorsal and ventral subdomains of the optic lobe (green). **e**, **f** Changes in N activity (NRE-dVenus, green) upon *Dl* RNAi under the control of *optix-Gal4*. Dl (magenta) and Lsc (blue). The signal intensities of NRE-dVenus, Dl, and Lsc within the white (control) and yellow boxes (*Dl* RNAi) are plotted in the right panels. Asterisks indicate the *optix-Gal4* positive areas outlined by white lines. **e** Partial *Dl* knockdown causes the fusion of the twin peaks of N activity (yellow box, arrows). **f** Strong *Dl* knockdown causes the loss of Dl expression and N activity (yellow box). **g** A schema illustrating *trans*-activation and *cis*-inhibition at the wavefront. **h** Expression level of full-length N (white) is reduced behind the wavefront (Lsc, blue). Signal intensities of N (black) and Lsc (blue) within the white box are plotted in the right panel. Arrows indicate the reduction of N expression level. **i**, **j** Expression level of full-length N (white) is upregulated in *Dl* mutant clones visualized by the absence of GFP (green in **i**, blue in **j**) behind the wavefront (Lsc, blue). **i** Signal intensities of N (black) within the dotted box are plotted in the right panel. Arrow indicates the upregulation of N expression level. **j** N (magenta) is accumulated along the cell membrane (Ecad, green) in *Dl* mutant clones. The dotted box is magnified in the right panel. Scale bars indicate 20 μm.

cells adjacent to the proneural wavefront. Thus, Dl expression at the wavefront is responsible for generating the first and second N activity peaks.

The intuitive interpretation of the above results is that the Dl expression at the wavefront activates N signaling in the adjacent NE cells (first peak) and NBs (second peak). N activity is downregulated at the wavefront between the twin peaks of N activity (Fig. 2g). However, the first N activity peak partially overlaps with Dl expression in vivo (Fig. 1a–d). This is most likely because the proneural wave progresses in a medial-to-lateral orientation. While N activity establishes the twin peaks, the location of the Dl-expressing wavefront cells may change and partially overlap with the first N activity.

**Delta triggers Notch degradation at the wavefront.** The above results suggest that the strong Dl expression at the wavefront triggers quick inactivation of N signaling. Since the expression level of N is downregulated behind the wavefront, the expression of Dl may trigger N degradation (Fig. 2h). If this is the case, the expression level of N should be upregulated when the function of Dl is compromised. As expected, we observed striking upregulation of the full-length N protein, which accumulated along the plasma membrane visualized by DE-cadherin (Ecad), when *Dl* mutant clones were generated at the wavefront (Fig. 2i: $n = 15/25$, 2j: $n = 14/20$). In contrast, *N* messenger RNA (mRNA) level was not affected in *Dl* mutant clones (Supplementary Figure 5). These results suggest that post-translational N degradation upon Dl expression is the basis of the nonlinear nature of *cis*-inhibition.

In our mathematical model, the expression level of full-length N protein is assumed to be constant, while the term $-k_n N$ represents passive degradation of N signaling, which is independent of Dl expression. Considering the above experimental results, full-length N should be actively degraded through Dl-dependent *cis*-inhibition, which is incorporated into the model as active downregulation of N signaling in a Dl-dependent manner (Fig. 1e–g). Our model allows N to be negative (Supplementary Figures 1 and 2), which might correspond to the presence of a repressor complex of Su(H) and Hairless (H)[8]. We only use the parameter settings with which *N* remains nonnegative in the following study.

**Rab7 and Rab4 differentially colocalize with Delta.** It has been reported that Dl and N expressed in the same cell form a complex[14,34]. Consistent observations are found between N and Serrate (Ser), the other transmembrane ligand of N[35]. The formation of the Dl–N complex in *cis* may be the cause of the rapid N degradation. It has been reported that protein degradation in late endosomes regulates N signaling[15–24]. However, these studies

did not focus on the process of *cis*-inhibition. In order to examine if the degradation machinery in late endosomes is responsible for *cis*-inhibition at the proneural wavefront, we asked whether N and Dl are transported through the intracellular trafficking pathways.

Rab family small GTPases play diverse roles in intracellular trafficking. We systematically screened the Rab-EYFP library strains in which all of the endogenous Rab family genes were tagged with EYFP[36]. Since N is downregulated at the wavefront, we initially compared Dl expression with Rab-EYFP distribution. We found that Rab4 and Rab7 are colocalized with Dl at the wavefront outside the nucleus and inside the membranous Ecad signals (Fig. 3a–f). These puncta are most likely in the cytoplasm. We confirmed the results by using anti-Rab7 antibody and found that Dl, Rab4, and Rab7 colocalize in the same puncta (Fig. 3h)[37]. The proximity ligation assay (PLA) also suggested that Rab7 forms a complex with Dl in vivo (Fig. 3g).

Since Rab7 and Rab4 are known to play key roles in late and recycling endosomes, respectively[38,39], we hypothesized that the Dl–N complex is transported to the Rab7-positive late endosomes and that Dl is recycled to the plasma membrane through the Rab4-positive recycling endosomes.

Interestingly, we found that Rab4 signals are found inside the Dl puncta, while Rab7 mainly accumulates on the surface of the Dl puncta at a higher magnification (Fig. 3i). Indeed, the colocalization index of Dl-Rab4 was greater compared with that of Dl-Rab7 (Fig. 3i, right panel), suggesting that Rab4-dependent Dl recycling is more dominant than Rab7-dependent Dl degradation. The less prominent colocalization of Dl with Rab7 may explain why Dl is not degraded in late endosomes.

Although N is downregulated at the wavefront, we occasionally observed minor colocalizations of N with Rab7 and Rab4 (Fig. 3j). The colocalization indices of N and Rab7/4 were significantly lower compared with those of Dl and Rab7/4 (Fig. 3k). Importantly, N colocalized with Rab7 more significantly compared with Rab4, suggesting that Rab7-dependent N degradation is more dominant than Rab4-dependent N recycling. Similarly, colocalization of Dl-GFP, which recapitulates Dl distribution pattern, and N was occasionally observed (Fig. 3l and Supplementary Figure 5c). These results support the hypothesis that N is mainly degraded in late endosomes at the wavefront in a Dl-dependent manner.

**ESCRT complex downregulates Notch activity at the wavefront.** When proteins are transported from early to late endosomes (or multivesicular bodies), Rab7 and the ESCRT complex (ESCRT-I–III) regulate protein degradation through the fusion of late endosomes with lysosomes[21,38]. Since Dl colocalizes with

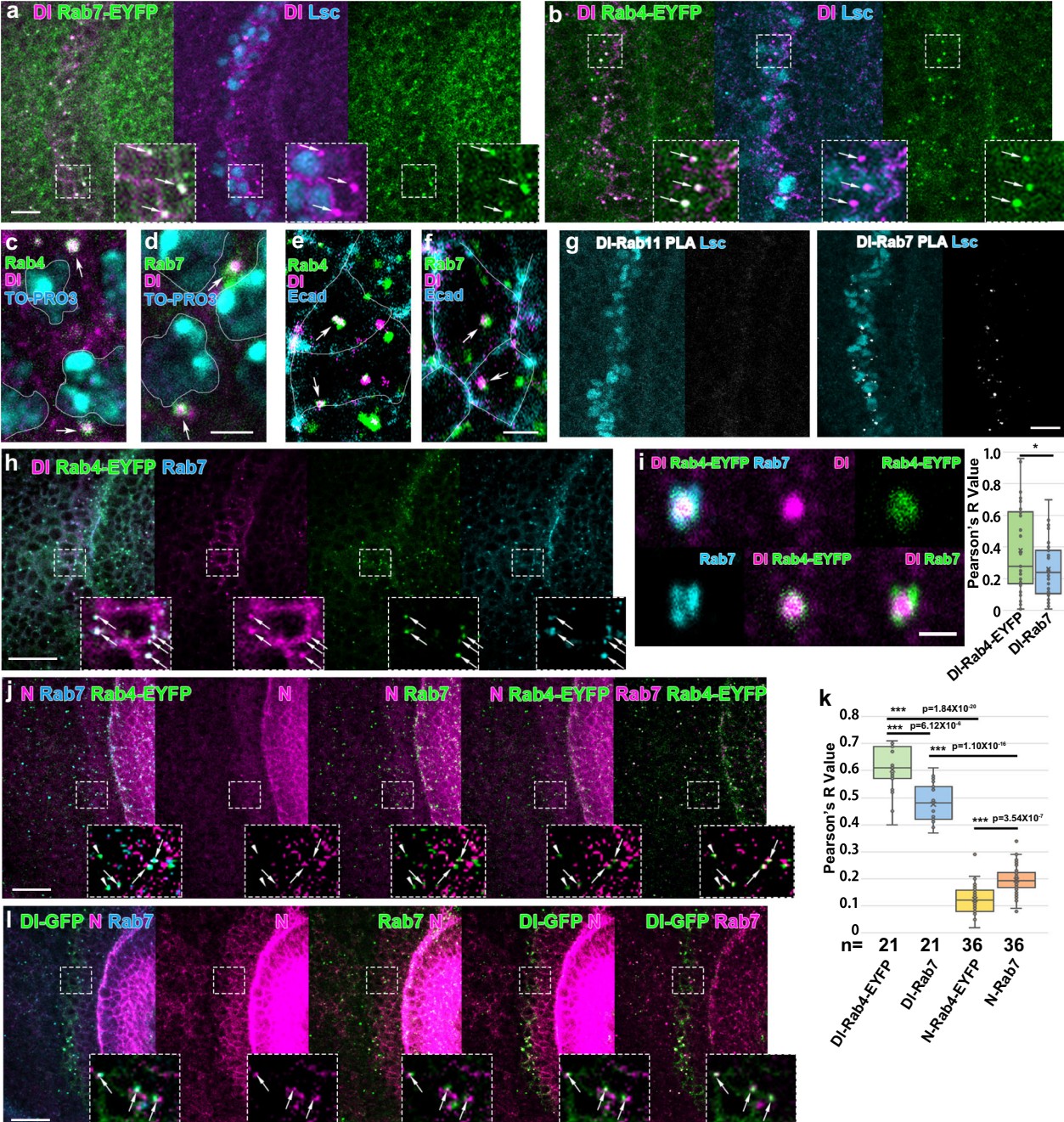

**Fig. 3 Delta and Notch differentially colocalize with Rab7 and Rab4. a**, **b** Dl (magenta) is colocalized with Rab7-EYFP (green in **a**) and Rab4-EYFP (green in **b**) at the wavefront (Lsc, blue) as indicated by arrows. **c–f** Dl (magenta) is colocalized with Rab7 and Rab4 outside the nucleus (TO-PRO3, blue) and inside the plasma membrane (Ecad, blue) as indicated by arrows. Nuclei and membranes are outlined by white lines. **g** Proximity ligation assay (PLA) demonstrates the interaction between Rab7 and Dl in vivo (white dots) at the wavefront (Lsc, blue). The interaction between Rab11 and Dl is not detected. **h** Dl (blue), Rab4-EYFP (green), and Rab7 (magenta) colocalize in the same puncta. **i** The colocalization of Dl, Rab7, and Rab4-EYFP is visualized in the left panels and quantified in the right panel. Pearson's $R$ values are compared for Dl with Rab4-EYFP and Rab7 ($p = 0.035$ (*$p < 0.05$), two-sided $t$ test, $n = 35$, number of quantified areas). **j** N is occasionally colocalized with Rab7 and Rab4-EYFP (arrows). **k** Pearson's $R$ values are compared for Dl and N with Rab4-EYFP and Rab7 ($p = 6.12 \times 10^{-6}$, $1.84 \times 10^{-20}$, $1.10 \times 10^{-16}$, and $3.54 \times 10^{-7}$ for Dl-Rab4/Dl-Rab7, Dl-Rab4/N-Rab4, Dl-Rab7/N-Rab7, and N-Rab4/N-Rab7, respectively (***$p < 0.001$), two-sided $t$ test, $n = 21$ (Dl) and 35 (N), number of quantified areas). **l** N is occasionally colocalized with Dl-GFP and Rab7 (arrows). Signals for N are enhanced. The dotted boxes are enlarged in the right bottom panels (**a**, **b**, **h**, **j**, **l**). Scale bars indicate 10 μm in (**a**, **b**, **g**, **h**, **j**, **l**), 2 μm in (**c–f**), and 1 μm in (**i**). Cross, mean; center line, median; box limits, upper and lower quartiles; whiskers, 1.5× interquartile range in the box plots (**i**, **k**).

Rab7 at the wavefront, we asked if the Dl–N complex is transported to late endosomes and N is degraded therein.

Many studies have reported the multiple roles of late endosomes in N signaling. When the function of late endosomes is compromised, the expression of Dl and N is upregulated. As a result, N signaling is ectopically activated[12,15–20,22–24]. On the other hand, it is also known that late endosome function is required for N activation. Indeed, N activity is downregulated

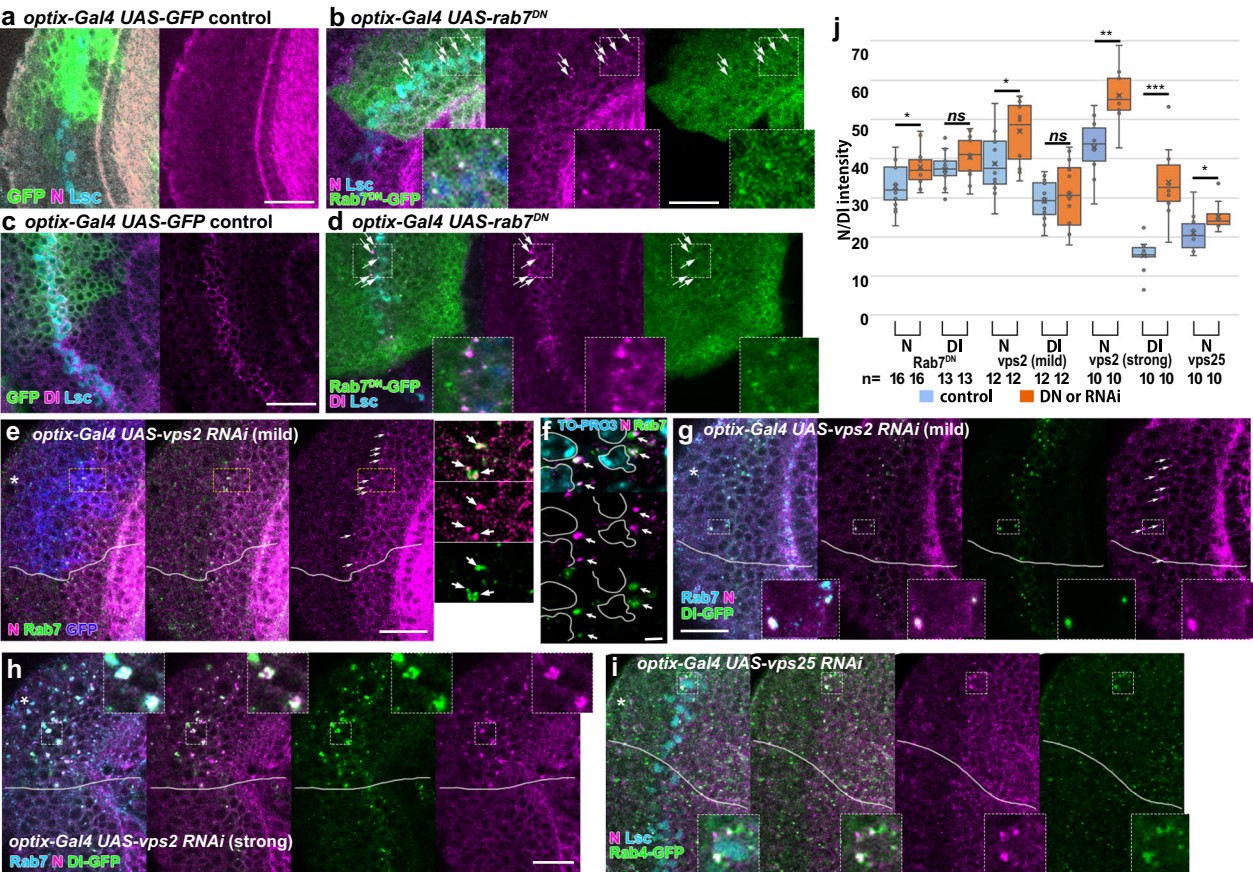

**Fig. 4 Late endosomes are responsible for Notch degradation at the wavefront. a–d** Expression of *UAS-rab7DN-GFP* under the control of *optix-Gal4* (green) induces ectopic puncta of N (magenta in **a**, **b**) and Dl (magenta in **c**, **d**) colocalized with Rab7DN-GFP (green, arrows) at the wavefront (Lsc, blue). The puncta of N and Dl are not detected in the controls (**a**, **c**). **e–h** vps2 RNAi under the control of *optix-Gal4* induces ectopic puncta of N (magenta) at the wavefront (arrows) in the areas indicated by asterisks (marked by *UAS-CD8GFP* expression in **e**). The N puncta are colocalized with Rab7 (green in **e**, **f**; blue in **g**, **h**) and Dl-GFP (green in **g**, **h**). Mild and strong phenotypes are shown in (**e–g**) and (**h**), respectively. The N puncta are found outside the nucleus (TO-PRO3, blue). **i** vps25 RNAi under the control of *optix-Gal4* induces ectopic puncta of N (magenta) that colocalize with Rab4-GFP (green) at the wavefront (arrows). The white dotted box area is enlarged in the right panels in (**e**, **g–i**). Scale bars indicate 20 μm in (**a–e**, **g–i**) and 2 μm in (**f**). **j** Quantification of the signal intensities of N and Dl ($p = 0.008$, 0.108, 0.180, 0.613, 0.0009, 0.0001, 0.042 (*$p < 0.05$, **$p < 0.01$, ***$p < 0.001$, n.s. not significant), two-sided $t$ test). The number of quantified areas is indicated. Cross, mean; center line, median; box limits, upper and lower quartiles; whiskers, 1.5× interquartile range.

when the function of the ESCRT protein is compromised in some experimental conditions[20]. Thus, late endosomes appear to be involved in multiple aspects of N signaling.

When we knocked down *rab7* by expressing the dominant-negative form of *rab7* or *rab7* RNAi, N protein level was slightly upregulated, showing punctate signals as visualized by N antibody (Figs. 4a, b, j and 5a, n). N and Dl colocalized with the Rab7DN puncta (Fig. 4a–d). However, the N activity reporter was not significantly affected when *rab7* was knocked down (see Fig. 5a).

Other late endosomal components may act redundantly with Rab7. We, therefore, focused on the functions of Vps family proteins that are involved in ESCRT complex function. When we knocked down *vps2*, a member of ESCRT-III, at the wavefront, N protein level was also upregulated, showing punctate signals (Fig. 4e, f). N and Dl colocalized in the Rab7- and Rab4-positive puncta (Fig. 4e–i).

We observed two distinct outcomes in N activity in the *vps2* RNAi background. In 40% of the cases ($n = 8/20$), N activity was specifically upregulated in cells situated between the twin peaks nearby the wavefront indicated by Lsc expression. The level of full-length N protein was slightly upregulated forming a dotted pattern. At the same time, the twin peaks were fused to form a

single peak (Figs. 4e, g, j and 5d–f, n and Supplementary Figure 6a), which is similar to the results of the partial knockdown of *Dl* shown above (Fig. 2e and Supplementary Figure 4a). Since the upregulation of the level of full-length N were not as prominent as those found in *Dl* mutant clones (Fig. 2i, j), there might be as yet unknown mechanisms that regulate N degradation and *cis*-inhibition. Importantly, Dl expression level was not significantly affected (Fig. 4g, j and Supplementary Figure 6b), suggesting that N activation and the formation of the single peak are not caused by changes in Dl expression.

Since EGF signaling indirectly influences N activity[3,27,40], the above phenotype may be the result of the changes in EGF activity. Since EGF activity as visualized by PntP1 staining was not significantly affected, the N activation phenotypes discussed above were not caused by indirect effects through EGF signaling (Supplementary Figure 7).

In the other 60% of the cases, Dl and full-length N proteins were widely upregulated (Figs. 4h, j and 5m and Supplementary Figure 6c, d; $n = 12/20$). Low-level N activity was observed in a wide area encompassing the wavefront (Fig. 5m and Supplementary Figure 6c), which may be related to the hyper-activation of N

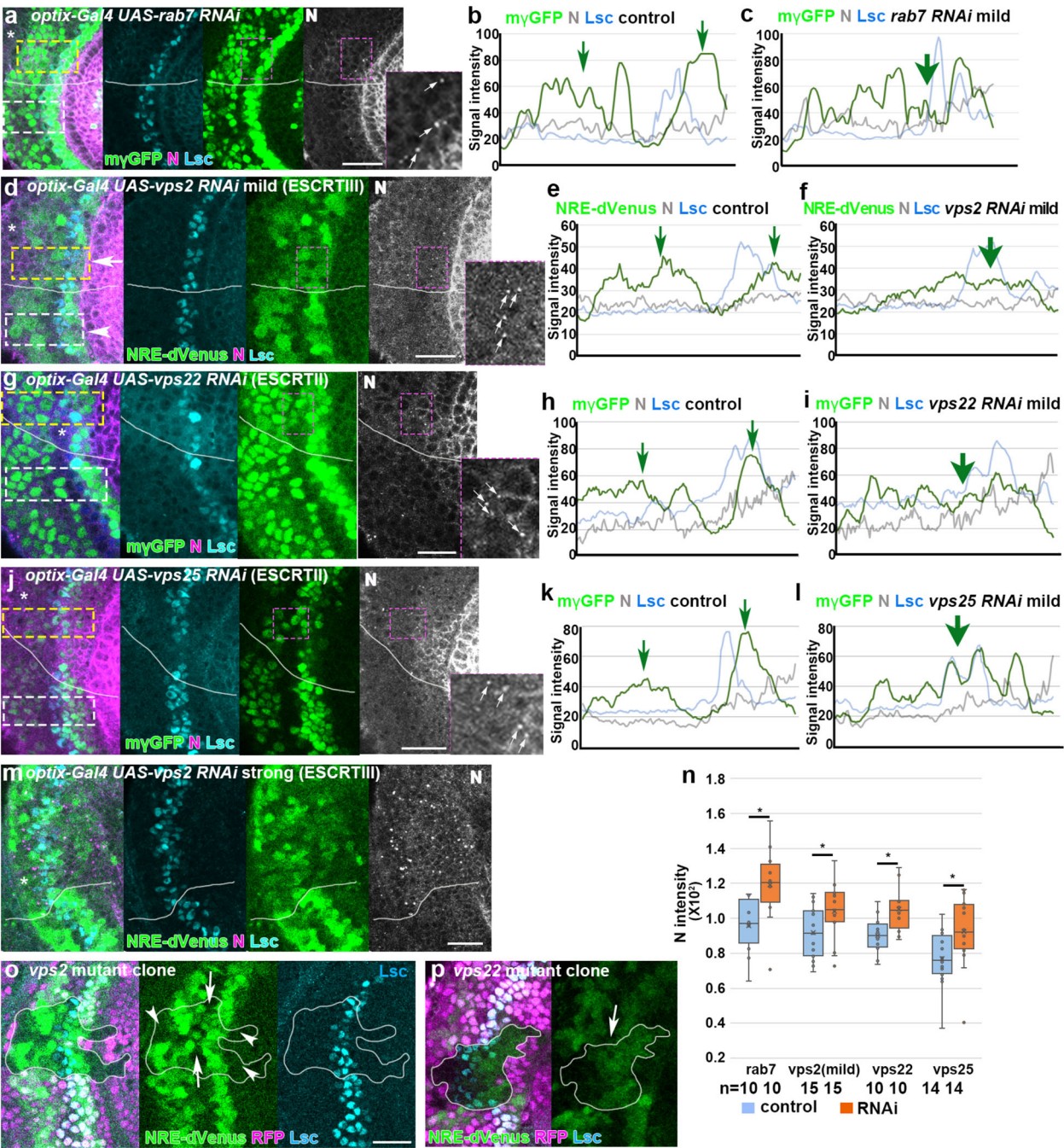

**Fig. 5 Late endosomes are responsible for Notch inactivation at the wavefront. a–l** RNAi knockdown of *rab7* (**a**), *vps2* (**d**), *vps22* (**g**), and *vps25* (**j**) under the control of *optix-Gal4* induces ectopic puncta of N (magenta or white, arrows) and fusion of the twin activation peaks of N (mγGFP in **a**, **g**, **j** and NRE-dVenus in **d**, green) at the wavefront (Lsc, blue). The magenta dotted boxes are magnified in the right panels showing the ectopic puncta of N. Asterisks indicate the *optix-Gal4* positive areas outlined by white lines. The signal intensities of N activity (green), N expression (gray), and Lsc (blue) within the white and yellow dotted boxes in (**a**, **d**, **g**, **j**) are plotted in (**b**, **e**, **h**, **k**; controls) and (**c**, **f**, **i**, **l**; RNAi), respectively. Green arrows indicate the N activity peaks. **m** *vps2* RNAi with a strong phenotype induces ectopic N puncta (magenta and white) and uniform N activity (NRE-dVenus, green) in a wide area (asterisk). Lsc (blue). **n** Quantification of N intensity ($p = 0.020$, $0.047$, $0.017$, $0.038$ (*$p < 0.05$), two-sided $t$ test). Numbers of quantified areas are indicated. Cross, mean; center line, median; box limits, upper and lower quartiles; whiskers, 1.5× interquartile range. **o**, **p** Fusion of the twin peaks of N activity (NRE-dVenus, green) in *vps2* and *vps22* mutant clones, respectively, visualized by the absence of RFP (magenta). Arrows indicate the upregulation of N activity between the twin peaks. Arrowheads indicate the absence of N activation in cells adjacent to the wavefront (Lsc, blue). Scale bars indicate 20 μm.

signaling and/or suppression of N activity in *vps2* mutant cells[15–18,20]. Since these phenotypes accompany smaller brain sizes compared with the brains showing the specific N activation phenotype discussed above, we assume that the mild RNAi effect caused the specific N activation between the twin peaks (Fig. 5d).

We repeated the same experiments for the other ESCRT complex genes and found essentially the same results (Fig. 5g–n and Supplementary Figure 8). Importantly, the specific N activation at the wavefront and the fusion of the twin peaks were reproducible in multiple RNAi conditions and mutant

clones of *vps2* and *vps22* (Fig. 5g–p and Supplementary Figure 8). Note that the twin peaks were partially fused, while ectopic N activation was not found in cells apart from the wavefront in the *vps2* and *vps22* mutant clones (Fig. 5o, p; $n = 9/15$ and 11/16). These results cannot be explained by the previously proposed roles of late endosomes that nonspecifically and uniformly degrade N and Dl. Thus, late endosomes play specific roles in N degradation that are essential for *cis*-inhibition at the proneural wavefront in addition to the roles in uniform nonspecific N degradation.

**Delta transport from late endosomes to recycling endosomes.** We hypothesize that the Dl–N complex is transported to late endosomes, where only N is degraded (Fig. 6o). We also hypothesize that Dl is released from late endosomes prior to its degradation and is recycled to the plasma membrane through recycling endosomes, because Rab4 colocalized with Dl more significantly compared with Rab7 (Fig. 3i).

When we knocked down *rab4* with RNAi, Dl expression was accumulated at the wavefront in a milder condition at 25 °C (Fig. 6a, b, m) and the colocalization of Dl with Rab7 was significantly increased (Fig. 6b). Interestingly, Dl expression was downregulated in a stronger RNAi condition at 30 °C (Fig. 6c, m). These results suggest that Dl is retained in Rab7-positive late endosomes and is degraded together with N when the function of recycling endosomes is eliminated. Furthermore, overexpression of *rab4*$^{DN}$ mimicked the effects of *rab4* RNAi in the milder condition, and aggregated distribution of Dl colocalized with Rab7 in the cytoplasm of cells at the wavefront (Supplementary Figure 9). Similar results were demonstrated in cells mutant for the components of recycling endosomes in the central brain[41].

It is known that late endosomes become acidic in the course of fusion with lysosomes, by which proteins are degraded[42]. The decrease in pH may trigger Dl dissociation from the Dl–N complex. Rabconnectin (Rbcn) is a family of proteins that regulate endocytic trafficking by regulating the assembly and activity of vacuolar-ATPase (V-ATPase), which is responsible for the acidification of intracellular compartments[43,44]. Thus, we focused on the roles of *Rbcn* and *vha68-2*, a member of V-ATPase genes, in this process. In *vha68-2* mutant clones, Dl was aggregated and colocalized with Rab7 (Fig. 6d, e). When *vha68-2* was knocked down, signals indicating acidic cellular components were significantly reduced as visualized by Lysotracker (Supplementary Figure 10c). At the same time, N degradation was blocked, and the level of N protein was upregulated at the wavefront (Fig. 6f–j and Supplementary Figure 10). In this condition, we observed ectopic colocalization of Dl and N in Rab7 and Rab4-positive puncta (Fig. 6h–j), suggesting that acidification of the endocytic pathway triggers Dl dissociation from the Dl–N complex and N degradation. These results are consistent with the impaired endosomal acidification and accumulation of N in late endosomes and lysosomes in *vha68-2* mutant clones in eye imaginal disc[45].

Based on the above results, we hypothesize that the Dl–N complex in Rab7-positive late endosomes is dissociated upon pH decrease. As a result, N is degraded in lysosomes, while Dl is recycled to the cell membrane through Rab4-positive recycling endosomes. Since pH increase upon *vha68-2* RNAi caused Dl colocalization with Rab7 and Rab4 (Fig. 6h, i), Dl–N complex may be localized to either late endosomes or recycling endosomes when acidification is compromised.

If Dl is transported from late to recycling endosomes, Dl should more strongly colocalize with Rab7 when *rab4* is knocked down together with *vha68-2*. To test this possibility, we compared colocalization of Dl with Rab7 in *vha68-2* RNAi and *vha68-2 rab4*

double RNAi backgrounds. Indeed, the colocalization of Dl with Rab7 was significantly increased by knocking down *rab4* together with *vha68-2* (Fig. 6h, k, l, n). These results support the model shown in Fig. 6o.

**The second Notch activity controls neurogenesis.** The first peak of N activity is responsible for regulating the speed of proneural wave propagation[3,27]. What is the biological function of the second peak? We carefully compared the pattern of N activity and genes that are specifically expressed in the NBs behind the wavefront. Interestingly, the expression of one of the temporal transcription factors, Klu, coincides with the second peak of N activity (Fig. 7a, l)[31,46].

So far, Hth, Klu, Ey, Slp, D, and Tll show sequential expression in the medulla NBs[30,31]. However, the mutual interactions between Hth, Klu, and Ey have not been identified thus far. Since Hth is upregulated in NEs prior to the arrival of the proneural wave and is continuously expressed in NEs and NBs (Fig. 7l), the mechanism that regulates Klu expression remains unclear. The N activity in the second peak may regulate the onset of Klu expression in NBs.

We initially examined the effect of the complete elimination of Dl function by generating *Dl*-null mutant clones in which the proneural wave is accelerated[27]. Since Klu expression persists for a long time following its induction at the second N activity peak, the premature NB differentiation would still show a persistent Klu expression behind the accelerated wavefront. However, Klu expression was eliminated in *Dl* mutant cells except for the cells situated along the boundary between the Dl-positive cells (Fig. 7b, $n = 16$). The residual Klu expression along the clone boundary may be due to the non-autonomous effect of N *trans*-activation. In contrast, the expression Hth, Ey, and Slp was not significantly affected in *Dl* mutant clones (Supplementary Figure 11a–c). These results suggest that N signaling is indeed necessary for Klu expression in NBs.

Since we do not have a technique that specifically inactivates the second N peak, we made use of the partial knockdown of *Dl* and *vps2*, resulting in the loss of the gap between the twin peaks and, consequently, the fusion of the twin peaks. Interestingly, Klu was precociously expressed in the newborn NBs under these conditions without significantly affecting the proneural wave progression (Fig. 7c, d), suggesting that the second N activity peak in NBs indeed triggers the expression of Klu. Note that Klu expression is not activated in the first peak of N activity in NEs (Fig. 7a, l). Klu expression may require additional genetic factors that are specific to NBs.

We previously demonstrated that Hth expression in the newborn NBs promotes the production of brain-specific homeobox (Bsh)-positive neurons that form the innermost concentric zone in the larval medulla (Fig. 7e, f, l)[47,48]. In slightly older NBs, Klu expression triggers the production of Runt (Run)-positive neurons, forming a concentric zone just outside the Bsh-positive neurons[31]. Consistent with the precocious Klu expression in NBs, we occasionally observed Run-positive neurons in an area inside the Bsh-positive neurons in *Dl* and *vps2* RNAi conditions (Fig. 7g, h, i). These defects were restricted to the dorsal part of the brain, most likely due to the stronger expression of *optix-Gal4* in the dorsal brain (Fig. 7e). The Klu expression in the newborn NBs, which only express Hth in the control background, might cause the production of Run-positive neurons earlier than Bsh-positive neurons, resulting in their abnormal distributions in the medulla (Fig. 7l). Note that Hth expression is widely found in NE and NB cells, and is not affected by Klu[31]. Therefore, the expression of Hth and production of Bsh-positive neurons should not be affected.

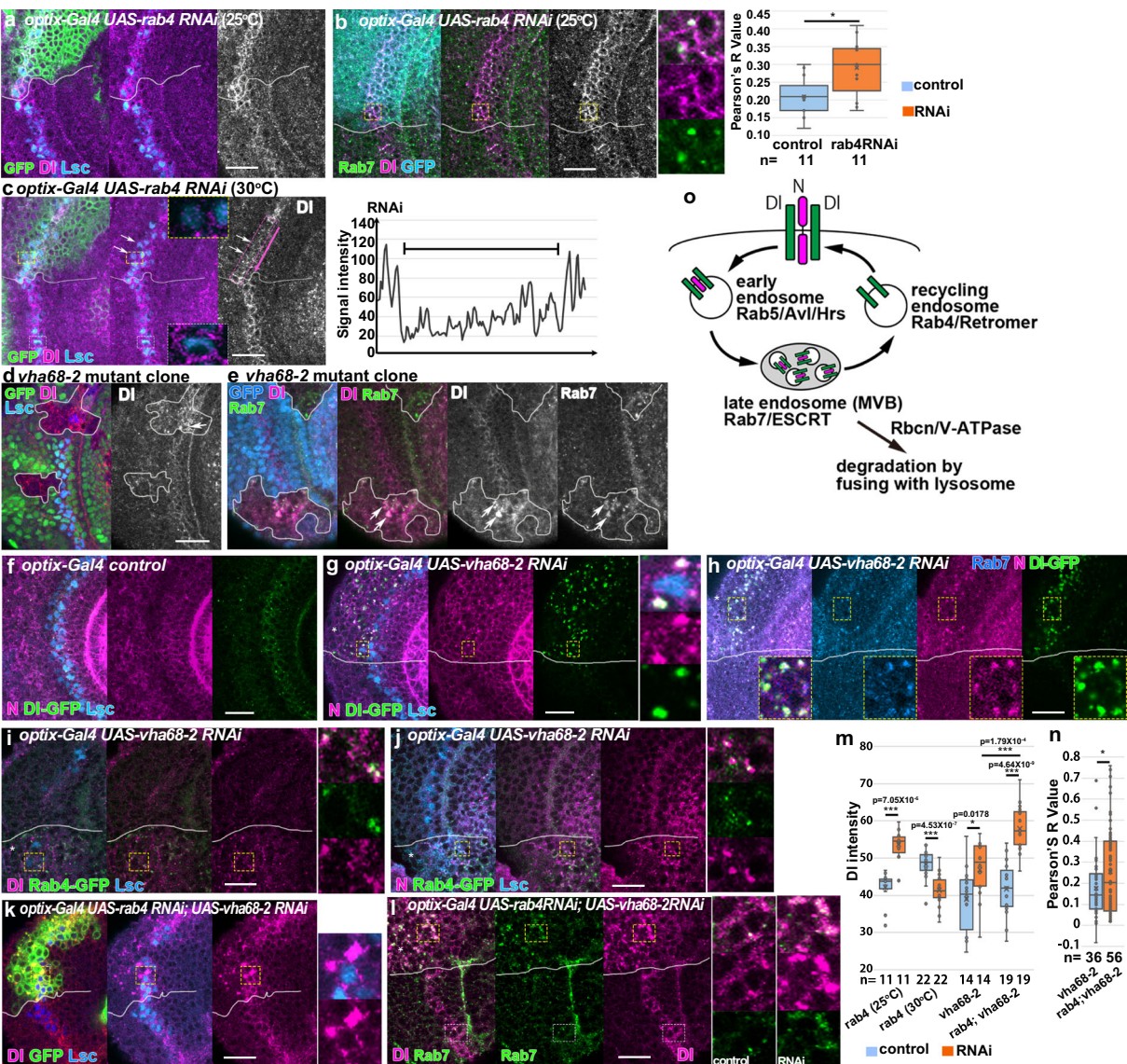

**Fig. 6 Delta transport from late endosomes to recycling endosomes. a**, **b** Dl aggregation (magenta or white) upon *rab4 RNAi* (*optix-Gal4*, green in **a**, blue in **b**) at 25 °C. Lsc, blue (**a**) and Rab7, green (**b**). Dl colocalizes with Rab7 (**b**). The white dotted box is magnified. The colocalization of Dl with Rab7 is quantified (*p* = 0.016 (**p* < 0.05), two-sided *t* test). **c** Loss of Dl expression (arrows) by *rab4 RNAi* (*optix-Gal4*, green) at 30 °C at the wavefront (Lsc, blue). The white dotted boxes are enlarged showing the decrease of Dl in the mutant area (yellow) compared with the control area (white). Signals in the magenta dotted box are plotted in the right panel. The magenta arrow indicates the orientation along the x-axis. **d**, **e** Dl is upregulated in *vha68-2* mutant clones (arrows) visualized by the absence of GFP (green in **d**, blue in **e**) at the wavefront (Lsc, blue in **a**). Ectopic Dl is accumulated in Rab7-positive structures (green in **e**). **f** Control distributions of N (magenta), Dl-GFP (green), and Lsc (blue). **g–j** *vha68-2* RNAi under the control of *optix-Gal4*. Distribution of ectopic N puncta (magenta in **g–j**), Dl-GFP (green in **g**, **h**), Dl (magenta in **i**), Rab7 (blue in **h**), and Rab4-GFP (green in **i**, **j**) is compared behind the wavefront (Lsc, blue in **g**, **i**, **j**). **k**, **l** *rab4 vha68-2* double RNAi under the control of *optix-Gal4* (GFP, green in **k**). Dl (magenta), Rab7 (green in **l**), and Lsc (blue in **k**). The yellow and white dotted boxes in RNAi and control areas, respectively, are magnified in the right panels. Asterisks indicate the *optix-Gal4*-positive areas outlined by white lines. **m** Dl intensity quantification in different RNAi conditions (*p* values are indicated inside the panel, **p* < 0.05, ****p* < 0.001, two-sided *t* test). **n** Quantification of the colocalization between Dl and Rab7 (*p* = 0.041 (**p* < 0.05), two-sided *t* test). **b**, **m**, **n** Numbers of quantified areas are indicated. Cross, mean; center line, median; box limits, upper and lower quartiles; whiskers, 1.5× interquartile range in the box plots. **o** Putative intracellular mechanisms that degrade N and recycle Dl. Scale bars, 20 μm.

We have demonstrated that Bsh-positive neurons give rise to Mi1 medulla neurons[47,48]. Similarly, the results of clonal analysis using *run-Gal4* demonstrate that Run-positive neurons differentiate into Mi4 and TmY16 neurons (Fig. 7j, k; *n* = 60 and 49, respectively). The neuronal-type TmY16 has not been documented based on its projection pattern in the medulla, lobula, and lobula plate (Dr. Kazunori Shinomiya, personal communication). Thus, the temporal regulation of the N dynamics at the proneural wavefront controls the temporal

pattern of neuronal-type specification through the expression of the temporal transcription factor Klu.

## Discussion

In this study, we incorporated nonlinearity in *cis*-inhibition to the mathematical model of the proneural wave and reproduced the twin activation peaks of N signaling at the wavefront as observed in vivo. The fast nonlinear dynamics of *cis*-inhibition compared

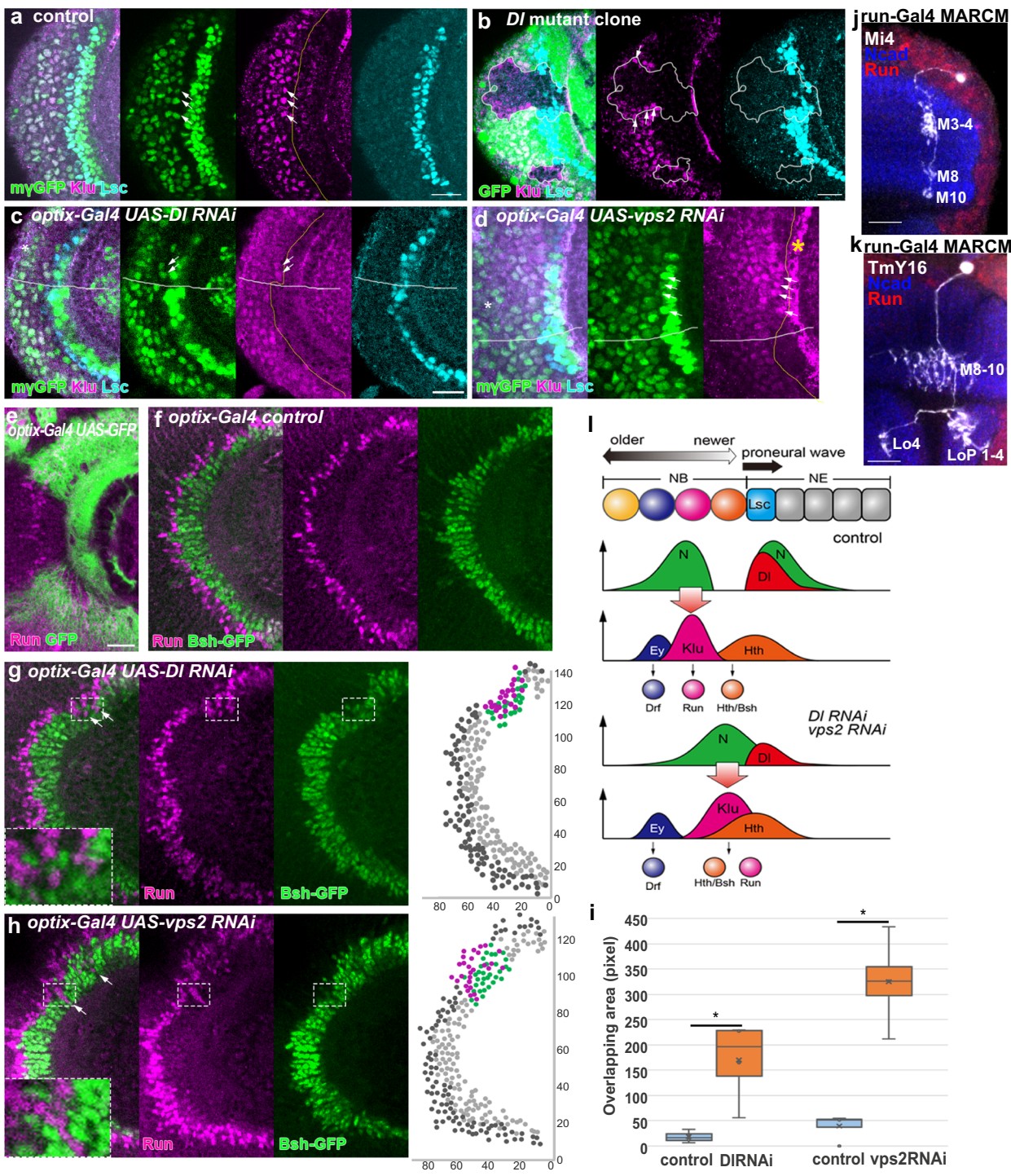

with those of the gradual kinetics of *trans*-activation is consistent with the results of the in vitro cell culture system[11]. The differential kinetics of *trans*-activation and *cis*-inhibition provide the rich dynamics of N activity that enable the formation of the twin activation peaks and the regulation of the temporal patterning of neurogenesis.

According to the previous literatures, the upregulation of Dl expression may induce the clustering of N and Dl[14,34,35], which leads to an acute suppression of N signal activity via *cis*-inhibition (Fig. 6o). Consistent with this idea, the partial knockdown of *Dl* caused an upregulation of N activity between the twin peaks and

their fusion, which reflects the failure in *cis*-inhibition. The expression level of full-length N is temporally downregulated behind the wavefront, which is derepressed in *Dl* mutant clones (Fig. 2h, i), suggesting that *cis*-inhibition is triggered by N degradation in response to Dl expression.

We essentially focused on the roles of intracellular trafficking in regulating N degradation and *cis*-inhibition (Fig. 6o). RNAi and mutant clones of ESCRT complex genes caused the fusion of the twin peaks of N activity, which is similar to the result of the partial *Dl* knockdown. We assume that the ESCRT complex regulates N degradation in the presence of high levels of Dl

**Fig. 7 The second Notch peak regulates the temporal patterning of neurogenesis. a** Klu expression (magenta) coincides with the second peak of N activity (mγGFP, green) as indicated by arrows behind the wavefront (Lsc, blue). **b** Klu expression (magenta) is eliminated in *Dl* mutant clones visualized by the absence of GFP (green), except for cells at the clone boundary (arrows). Lsc visualizes the accelerated wavefront inside the *Dl* mutant clones (blue). **c, d** Fusion of the twin peaks of N activity (mγGFP, green) and precocious Klu expression (magenta) indicated by arrows upon partial knockdown of *Dl* (**c**) and *vps2* (**d**). Asterisks indicate the *optix-Gal4*-positive areas outlined by white lines. Yellow lines outline Klu-expressing NBs. The wavefronts are not accelerated (Lsc, blue). **e** *optix-Gal4* is expressed in the dorsal (top) and ventral (bottom) subdomains of the optic lobe (GFP, green). GFP signals are stronger in the dorsal subdomain. Run (magenta). **f** Cells expressing Bsh-GFP (green) and Run (magenta) form the concentric zones that are complementary with each other. **g, h** Ectopic Run-positive cells (magenta) are found within the concentric zone expressing Bsh-GFP (green) upon partial knockdown of *Dl* and *vps2* as indicated by arrows. Positions of Run and Bsh-positive cells are plotted in magenta and green within the defective areas and in black and gray in the other areas, respectively, in the right panels. **i** Overlaps between Run- and Bsh-positive areas are quantified in *Dl* and *vps* RNAi backgrounds ($p = 0.034$, 0.009 (*$p < 0.05$, **$p < 0.01$), two-sided *t* test, $n = 4$, number of brain samples). Cross, mean; center line, median; box limits, upper and lower quartiles; whiskers, 1.5× interquartile range. **j, k** Morphologies of Run-positive medulla neurons, Mi4 (**j**) and TmY16 (**k**) visualized by *run-Gal4 UAS-CD8GFP* MARCM clones (white). Run (red) and Ncad (blue). Layers in the medulla (M), lobula (Lo), and lobula plate (Lop). **l** Schematics of the temporal patterning of neurogenesis through N activity and Klu expression. Hth- and Klu-positive NBs produce Bsh- and Run-positive neurons, respectively. Fusion of the twin peaks of Notch activity leads to precocious Klu expression, which precociously produces Run-positive neurons. Scale bars, 20 µm.

expression. Since the expression level of full-length N was only partially upregulated, forming a dotted pattern by knocking down *rab7* and ESCRT genes, there might be as yet unknown mechanisms that regulate N degradation (Fig. 5, Supplementary Figure 6).

We assumed that the clustering of N and Dl is the trigger of N degradation. However, Dl is not degraded at the wavefront, as evident from its strong membrane localization, despite its localization in Rab7-positive late endosomes. Since the colocalization of Dl with Rab4 is more prominent than that with Rab7, recycling of Dl to the plasma membrane may be dominant compared with its degradation (Fig. 3i). Indeed, a mild knockdown of *rab4* caused Dl accumulation in Rab7-positive late endosomes, while its severe knockdown induced Dl degradation (Fig. 6a–c). Colocalization of Dl with Rab7 upon blocking acidification in late endosomes was further enhanced by the simultaneous knockdown of *vha68-2* and *rab4* (Fig. 6l, n). These results suggest that Dl is dissociated from the Dl–N complex in late endosomes, and is recycled to the plasma membrane through recycling endosomes (Fig. 6o).

The mathematical models in Fig. 1 do not explicitly consider the degradation of N protein upon its *cis*-interaction with Dl. We further improved the model by considering full-length N (*F*) and active form of N (*S*). In a wide range of parameter settings, this model reproduces the formation of the twin peaks of N activity, fast degradation, and gradual recovery of the expression level of full-length N (Supplementary Figure 12). Although Dl function is thought to be inhibited when Dl and N interact in *cis*[11], we did not include this reaction in the model, because we do not have any observation that suggests *cis*-inhibition of Dl in the course of the proneural wave progression.

Furthermore, we demonstrated a role of N signaling in regulating the temporal patterning of neural progenitor cells (NPCs) by focusing on the transition of the temporal transcription factor Klu, whose expression is upregulated in the medulla NBs[31]. We showed that Dl remotely regulates Klu expression behind the proneural wave through the second peak of N activity. Thus, fine temporal regulation of N activity through *cis*-inhibition plays essential roles in the temporal patterning of neurogenesis in the fly brain.

In the NPCs of the developing cerebral cortex, the temporal dynamics of N activity also play important roles in the temporal patterning of neurogenesis and gliogenesis[49]. In this process, the basic helix–loop–helix transcription factors show oscillatory expression in NPCs and N signaling appears to perform lateral inhibitory feedback during NPC differentiation. The roles of *cis*-inhibition in this process remain largely elusive. It will be interesting to see how the molecular mechanisms revealed in the current study are conserved in a wide variety of developmental processes.

## Methods

**Mathematical modeling and numerical simulation.** The differential equations were calculated using the explicit finite difference method with the zero flux boundary condition as described previously[3]. The mesh size of the square grid model is equal to the cell size ($dx = 2$, $25 \times 25$ cells), and the time step size is 0.01.

*Four-variable model*: The model contains four equations and four variables. $E$ is a composite variable for EGF ligand concentration and EGF signaling. $N$ represents N signal activity. $D$ and $A$ represent expression levels of Dl and AS-C, respectively. $d_e$ is the diffusion coefficient of EGF. $d_t$ and $d_c$ represent magnitudes of the *trans*-activation and *cis*-inhibition, respectively. $i, l$ and $j, m$ are integers indicating the location of a cell along the $x$ and $y$ axes, respectively. $l$ and $m$ indicate the location of four cells adjacent to a cell indicated by $i$ and $j$. The initial conditions for $A$, $E$, and $D$ in the anterior-most cells ($x = 1, 2, 3$) are $A = 0.90, 0.31$, and 0.02; $E = 0.054, 0.021$, and 0.0016; and $D = 0.062, 0.021$, and 0.0013, respectively, to stabilize the proneural wave progression. The initial condition of $N$ is 0 in all cells.

$$\frac{dE_{i,j}}{dt} = d_e \Delta E_{i,j} - k_e E_{i,j} + a_e A_{i,j}\left(1 - A_{i,j}\right) \quad (1-1)$$

$$\frac{dN_{i,j}}{dt} = -k_n N_{i,j} + d_t \sum_{l,m \in \Lambda_{i,j}} D_{l,m} - d_c D_{i,j} \quad (1-2)$$

$$\frac{dD_{i,j}}{dt} = -k_d D_{i,j} + a_d A_{i,j}\left(1 - A_{i,j}\right) \quad (1-3)$$

$$\frac{dA_{i,j}}{dt} = e_a\left(1 - A_{i,j}\right)\max\left\{E_{i,j} - N_{i,j}, 0\right\} \quad (1-4)$$

The diffusion of $E$ was calculated as follows: $d_e \Delta E_{i,j} = d_e\left(E_{i+1,j} + E_{i-1,j} + E_{i,j-1} + E_{i,j-1} - 4 E_{i,j}\right)/dx^2$. $k_e$, $k_n$, and $k_d$ are passive degradation rates of EGF, N, and Dl, respectively. $a_e$ and $a_d$ indicate EGF and Dl regulation by AS-C, respectively. When $A = 1$, the cells are fully differentiated as NBs. $e_a$ reflects the speed of differentiation under the control of EGF and N, and is set to 100. The other parameters are set to 1 unless otherwise noted.

*Linear cis-inhibition*: In the previous model, the *trans*-activation and *cis*-inhibition are linear (1-2). $k_n = 1$, $d_t = 0.25$, and $d_c = 0.1$ (Fig. 1e). In this study, (1-2) is replaced with the following.

*Step function*: (1-2) is replaced with (2-1) and (2-2). *cis*-inhibition does not occur when $D$ is below the threshold ($e_c$), but quickly induces N inactivation when it exceeds the threshold. $k_n = 1$, $d_t = 0.25$, $d_c = 0.1$, and $e_c = 0.07$ (Figs. 1f and 2b). Magnitude ($d_c$) and threshold ($e_c$) of *cis*-inhibition are changed in the range 0.07–0.15 and 0.04–0.08, respectively (Supplementary Figure 1).

$$\frac{dN_{i,j}}{dt} = -k_n N_{i,j} + d_t \sum_{l,m \in \Lambda_{i,j}} D_{l,m} - d_c H(D_{i,j}) \quad (2-1)$$

$$H(x) = \begin{cases} 1 \ (x \geq e_c) \\ 0 \ (x < e_c) \end{cases} \quad (2-2)$$

*Hill function*: (1-2) is replaced with (3-1). Hill's coefficient ($n_t$ and $n_c$) and activation coefficient ($k_t$ and $k_c$) specify the kinetics of the reaction. $k_n = 1$, $d_t = 0.25$, $d_c = 0.1$, $n_t = 1$, $k_t = 1$, $n_c = 5$, and $k_c = 0.09$ (Figs. 1g and 2c). Hill's coefficient ($n_c$) and activation coefficient ($k_c$) of *cis*-inhibition are changed in the

range 2–7 and 0.07–0.11, respectively (Supplementary Figure 2).

$$\frac{dN_{i,j}}{dt} = -k_n N_{i,j} + \frac{d_t \left( \sum_{l,m \in \Lambda_{i,j}} D_{l,m} \right)^{n_t}}{k_t^{n_t} + \left( \sum_{l,m \in \Lambda_{i,j}} D_{l,m} \right)^{n_t}} - \frac{d_c D_{i,j}^{n_c}}{k_c^{n_c} + D_{i,j}^{n_c}} \quad (3-1)$$

*Five-variable model*: (1-2) is replaced with (4-1) and (4-2). $F$ and $S$ represent the expression level of full-length N and N signal activity, respectively, whose initial conditions are 1 and 0 in all cells. $F_{max}$, $n$, and $k_f$ represent maximum expression level, recovery rate, and spontaneous degradation rate of full-length N, respectively. $d_t$ and $d_c$ represent magnitudes of the *trans*-activation and *cis*-inhibition, respectively. Hill's coefficient ($n_t$ and $n_c$) and activation coefficient ($K_1$ and $K_2$) specify the kinetics of the reaction. $k_t$ represents the degradation rate of full-length N upon *trans*-activation. $k_f$ represents passive degradation rate of active form for N. $n = 1$, $F_{max} = 1$, $k_f = 0.1$, $d_c = 5$, $n_t = 1$, $K_1 = 0.3$, $k_t = 0.1$, $d_t = 0.25$, and $k_s = 2$. Hill's coefficient ($n_c$) and activation coefficient ($K_2$) for Hill function in *cis*-inhibition are changed in the range 1–7 and 0.05–0.3, respectively (Supplementary Figure 12).

$$\frac{dF_{i,j}}{dt} = n \left( F_{max} - F_{i,j} \right) - k_f F_{i,j} - \frac{d_c F_{i,j} D_{i,j}^{n_c}}{K_2^{n_c} + D_{i,j}^{n_c}} - \frac{k_t F_{i,j} \left( \sum_{l,m \in \Lambda_{i,j}} D_{l,m} \right)^{n_t}}{K_1^{n_t} + \left( \sum_{l,m \in \Lambda_{i,j}} D_{l,m} \right)^{n_t}} \quad (4-1)$$

$$\frac{dS_{i,j}}{dt} = \frac{d_t F_{i,j} \left( \sum_{l,m \in \Lambda_{i,j}} D_{l,m} \right)^{n_t}}{K_1^{n_t} + \left( \sum_{l,m \in \Lambda_{i,j}} D_{l,m} \right)^{n_t}} - k_s S_{i,j} \quad (4-2)$$

Two- and one-dimensional plots in Figs. 1e–g, 2b, c, and Supplementary Figures 1, 2, and 12 are the snapshots when the state of differentiation of the central cell exceeds 0.5 ($A_{13,13} > 0.5$). The source codes for the numerical simulations will be deposited to a public repository service.

**Fly strains**. Fly strains were maintained on standard *Drosophila* medium at 25 °C. *rab4* RNAi was performed at 25 and 30 °C. The following mutant and transgenic flies were used: *Dl^RevF10^ FRT82B*, *UAS-Dl^3-1^*, *UAS-Dl RNAi* (strong: BDSC36784; mild: V52188), *Dl-GFP* (BDSC59819), *rab7-EYFP* (BDSC62545L), *rab4-EYFP* (BDSC62542), *UAS-rab7^DN^* (BDSC9778), *UAS-rab7 RNAi* (BDSC27051, VDRC40338), *UAS-rab4^DN^* (BDSC9768, BDSC9769), *vps2^pp6^ FRT82B*, *vps22^NN31^ FRT82B*, *UAS-vps2 RNAi* (BDSC38995), *UAS-vps 22RNAi* (BDSC38289), *UAS-vps23 RNAi* (BDSC38306), *UAS-vps25 RNAi* (BDSC26286), *UAS-vps24 RNAi* (BDSC38281), *UAS-vps32 RNAi* (BDSC38305), *UAS-vps20 RNAi* (BDSC40894), *UAS-vps36 RNAi* (BDSC38286), *UAS-vps37B RNAi* (BDSC44010), *UAS-vha68-2 RNAi* (BDSC34582), *vha68-2^R6^ FRT40A* (BDSC39621), *UAS-rbcn3A RNAi* (BDSC34612), *optix-Gal4* (NP2631), *Ay-Gal4*[50], *UAS-CD8GFP*, *UAS-GFP*, *hs-flp*, *ubi-GFP FRT82B*, and *ubi-RFP FRT82B*. N activity was visualized by *E(spl)-myGFP* (*myGFP*) and *NRE-dVenus*[32,33].

**Clonal analysis**. *Dl* overexpression clones were generated by crossing *hs-flp; Ay-Gal4 UAS-GFP* strain with *UAS-Dl^3-1^* and applying 15 min heat shock at 34 °C. *Dl* mutant clones were generated by crossing *hs-flp; ubi-GFP FRT82B* strain with *Dl^RevF10^ FRT82B* and applying 50 min heat shock at 37 °C. *vps* mutant clones were generated by crossing *hs-flp; myGFP; ubi-RFP FRT82B or hs-flp; NRE-dVenus; ubi-RFP FRT82B* with *vps FRT82B* and applying 50 min heat shock at 37 °C. *run-Gal4* MARCM clones were generated by crossing *run-Gal4 UAS-CD8GFP FRT19F; FRT40A* strain with *hs-flp; tubP-Gal80 FRT40A* and applying 15 min heat shock at 34 °C.

**Generation of *run-Gal4* strain**. The second exon containing the translational start site of *run* was replaced by the fragment containing *attP* site and *GMR-white* via the homologous recombination with the homology arms for *run* (runLp and runRp)[51] in the presence of *FRT19F* (Kyoto125719). The Gal4 containing fragment was integrated into the *attP* site in the *run* locus by co-injecting PhiC31 plasmid and *run-Gal4/pGEattB*, which contains *attB* site, the fragment containing the translational start site of *run* (run5pri) and *GMR-white*. The expression pattern of *run-Gal4* was verified by expressing *UAS-GFP* and co-staining with anti-Run antibody. runLp (2.0 kb), runRp (2.2 kb, and run5pri (153 bp) were amplified using the following PCR primers: runLp5_*Not*I (catgcggccgcccaagtatgacacttccgcatc), runLp3_*Kpn*I (catggtacccttttatcgggggtcacttggaa), runRp5_*Asc*I (catggcgcgcccgg-gagccaagaagtaagcaaa), runRp3_*Xho*I (catctcgagggccaactgtgataggaagttc), runGal5-*Not* (catgcggccgcttccaagtgaccccccgataaag), and runGal3-*Bam* (catg-gatcctgtgttgttggccaccatcgtt). Underlined sequences are homologous to the genomic sequences.

**Histochemistry**. Immunohistochemistry was performed as described below[48]. Details are available upon request. Brains were dissected in phosphate-buffered saline (PBS), transferred to ice-cold 0.8% formaldehyde/PBS solution, and fixed in 4% formaldehyde/PBS at room temperature for 30–60 min. The brains were washed in PBT (0.3% Triton X in PBS) and blocked in 5–10% normal serum/PBT solution at room temperature for 30–60 min. Primary antibody reaction was performed in a solution containing primary antibodies and 1% normal serum in PBT

at 4 °C overnight. The brains were washed in PBT. Secondary antibody reaction was performed in a solution containing secondary antibodies and 1% normal serum in PBT at 4 °C overnight. After washing in PBT and PBS, the brains were mounted in VECTASHIELD (Vector Laboratories).

The following primary antibodies were used: guinea pig anti-Lsc (1:1200), mouse anti-Dl (1:20; DSHB C594.9B), mouse anti-N (1:20; DSHB C17.9C6), rat anti-Ecad (1:20; DSHB DCAD2), rabbit anti-Klu (1:1000; Xiaohang Yang, Singapore), guinea pig anti-Run (1:500; Asian Distribution Center for Segmentation Antibodies, Mishima, Japan), and rabbit anti-Rab7 (1:600; Akira Nakamura, Kumamoto University, Japan). The following secondary antibodies were used: anti-mouse Cy3 (1:200; Jackson ImmunoResearch 715-165-151), anti-mouse Cy5 (1:200; Jackson ImmunoResearch 715-175-151), anti-guinea pig Alexa647 (1:200; Jackson ImmunoResearch 712-605-150), and anti-rabbit Alexa546 (1:200; Invitrogen A-11035). Confocal images were obtained by Zeiss LSM880 and processed using ZEN, Fiji, and Adobe Photoshop. Signal intensity was quantified within the indicated rectangle areas by ImageJ.

**In situ hybridization**. In situ hybridization was performed as briefly described below. Details are available upon request. Larval brains were dissected in ice-cold PBS, transferred to 4% formaldehyde/PBS solution, and fixed at 4 °C overnight. The formaldehyde solution was removed, and the brains were washed with PBS and 70% ethanol. Incubation at room temperature for 2–5 min after replacing the solution with Wash Buffer A (100 μl Stellaris RNA FISH Wash Buffer A, 300 μl nuclease-free water, 50 μl deionized formamide). Replacing the solution with Hybridization Buffer (90 μl Hybridization Buffer mix with 2 μl Stellaris RNA FISH probe designed for exon 6 of *N* gene labeled with Quasar 570), the brains were incubated at 37 °C for 8 h in the dark. Incubation at 37 °C for 30 min after replacing the solution with Wash Buffer A. Replacing the solution with TO-PRO3/PBS, the brains were incubated at 37 °C for 30 min. After washing in PBS, the brains were mounted in 80% glycerol in PBS.

**Duolink PLA**. PLA was performed using Duolink (Sigma-Aldrich). The dissected fly brains were fixed in 4% formaldehyde in PBT (0.3% Triton X in PBS). After finishing the Lsc immunostaining, the brains were incubated with Duolink blocking solution for 60 min at 37 °C and in the Duolink antibody diluent containing mouse anti-Dl and rabbit anti-Rab7 antibodies for overnight at 4 °C. The brains were washed in Duolink Wash Buffer A three times at room temperature, incubated in a solution containing PLA PLUS anti-mouse and PLA MINUS anti-rabbit probes for 120 min at 37 °C, washed in Duolink Wash Buffer A, and incubated with Duolink ligation stock for 60 min at 37 °C. Then, the brains were washed in Wash Buffer A and incubated with Duolink polymerase in Duolink amplification stock for 100 min at 37 °C. Finally, the brains were washed in Wash Buffer B and mounted in Duolink in situ mounting medium with DAPI (4′,6-diamidino-2-phenylindole).

**LysoTracker staining**. Fly brains were dissected in cold S2 medium and incubated in PBS containing LysoTrancker Red DND99 (1:1000, Thermo Fisher) for 30 min at room temperature. The brains were washed in PBS and fixed in 4% formaldehyde in PBS. After washing in PBS, the brains were mounted in VECTA-SHIELD (Vector Laboratories).

**Statistics and reproducibility**. For quantification and statistical analysis, distinct brain areas or samples were measured and analyzed as indicated below. Two-sided $t$ test was used for the statistical test. Image intensities were not artificially processed, except as otherwise noted. When statistics were not applicable, experiments were independently repeated at least three times with similar results.

**Image quantification**. Signal intensity was quantified within the indicated rectangle areas using Fiji (Figs. 1c–d, 2e–i, 4, 5, and 6f–n). Coloc 2 function of Fiji was used to calculate the Pearson's $R$ value to quantify the colocalization between two different signals (Figs. 3i, k and 6b, o).

The $xy$ coordinates of cells were obtained by manually selecting individual cells expressing Run or Bsh (Fig. 7g, h). The $x$ coordinate of the origin was calculated as the average of cell locations along the $x$-axis. The $y$ coordinate of the origin was determined so that the standard deviation of $R$, the distance from the origin to cells, is minimized. $\Delta\theta$, the angle of mutant area from the origin, was determined based on the mutant area of Bsh-positive cells. The same $\Delta\theta$ was used to determine the control area to be analyzed at the dorsal–ventral boundary where *optix-Gal4* is not expressed. Sizes of the intersection between Run- and Bsh-positive areas in the control and mutant areas were calculated using "convhull," "polyshape," and "intersects" functions of MATLAB (Fig. 7i).

**Ethical approval**. We have complied with all relevant ethical regulations for animal testing and research. This study did not require an ethical approval.

**Reporting summary**. Further information on research design is available in the Nature Research Reporting Summary linked to this article.

## Data availability

The authors declare that the data supporting the findings of this study are available within the paper, or available upon request. Source data are provided with this paper.

## Code availability

The source codes for the numerical simulations and image quantifications are deposited in a public repository service (https://github.com/satouma7/TwinPeaksCode).

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

## Acknowledgements
We thank members of Sato lab for supporting fly work, T. Kawauchi, K. Matsuno for helpful discussion, S. Bray, A. Nakamura, J. Skeath, Bloomington *Drosophila* Stock Center, Vienna *Drosophila* Resource Center, *Drosophila* Genetic Resource Center, Kyoto, Asian Distribution Center for Segmentation Antibodies, Mishima, and Developmental Studies Hybridoma Bank for flies and antibodies. This work was supported by CREST from JST (JPMJCR14D3 to M.S., S.-I.E. and JPMJCR15D2 to M.N.), Grant-in-Aid for Scientific Research (B), (C), Grant-in-Aid for Scientific Research on Innovative Areas and Grant-in-Aid for Early-Career Scientists from MEXT (17H03542, 17H05739, 17H05761, and 19H04771 to M.S., 19K06674, 19H04956, and 20H05030 to T.Y., and JP20K14364 to Y.T.), Takeda Science Foundation (to M.S. and T.Y.), Cooperative Research of "Network Joint Research Center for Materials and Devices" (to M.S.).

## Author contributions
M.W. and M.S. conceived and designed the experiments. X.H., C.L., R.T., T.Y., and M.S. performed experiments. M.W. and M.S. acquired, analyzed, and interpreted the data. S.-I. E., M.N., Y.T., and M.S. formulated the mathematical models. M.W. and M.S. wrote the manuscript.

## Competing interests
The authors declare no competing interests.
