## [Peer Review File · Nature Communications]

REVIEWER COMMENTS

Reviewer #1 (Remarks to the Author):

In the fly optic lobe, a wave of differentiation regulates the transformation of neuroepithelial cells into dividing neuroprogenitors (neuroblasts in the fly), that over time produce distinct neuronal progeny as they transit through a transcription factor temporal series. This cell fate transition resembles the transition of neuroepithelial to radial glial cells in the developing cerebral cortex.

Several studies (including those from the senior author) have shown how this transition involves a complex regulation of Notch signaling. However, unlike what is typical for Notch signaling, the spatial activation of the pathway does not form a salt and pepper pattern. Instead, a wave of Notch activity propagates across the neuroepithelium to regulate its differentiation into neuroblasts. However, Notch has to be downregulated for the differentiation to occur and newborn neuroblasts do not show Notch activity. A second peak of Notch activity is observed as neuroblasts age. Thus, two peaks of Notch activity are under tight spatial and temporal control. The authors have previously published several exciting articles that provide a convincing model explaining how the first wave of Notch is formed and propagates across the neuroepithelium to regulate neuroblast production. However, their previous model could not explain the formation of the second wave of Notch that forms in older neuroblasts.

Here, the authors revisit their previous model. They provide evidence that this pattern of temporal Notch activation is dependent on cis-inhibition and relies on intracellular trafficking of Notch and Delta that leads to Notch degradation in late endosomes. They further show that the second wave of Notch activity in neuroblasts is required for the expression of the temporal factor Klu,

This paper provides an important contribution in the understanding of how neurogenesis and neural fate specification are orchestrated in the developing brain. However, I have some concerns regarding the data and their interpretation. In addition, the paper requires textual revisions, as it is often difficult to understand the points the authors are trying to raise. Therefore, the manuscript cannot be published before the following points are addressed.

My first recommendation is that the authors revise the text and work hard at writing clear and concise descriptions of the experiments and of the conclusions. For example, the abstract never explains what the "twin peaks" are, which makes the question confusing for the reader. Furthermore, although the authors mention that they have a prior mathematical model explaining the kinetics of the proneural wave, they never mention that this model utilizes Achaete-Scute and EGF signaling to model the wave, and the figure legends/text never mention what A and E stand for (leaving the reader to refer to the methods to decipher things). In addition to these points, other points might have been missed because of the unclear writing (see below).

Major points

1. For all manipulations in which the two waves of Notch activity are lost, the authors' interpretation is that the two waves are fused. A close look at their model (Figure 1g) shows this is not the case. Strong DI expression creates a decrease in Notch activity which is at the origin of the two waves. A reduction of its expression leads to reduced cis-inhibition: As a consequence, Notch is never downregulated and its expression is maintained from NE to DI expressing cells to aging neuroblasts.
2. "We could not detect co-localization of N with Rab7 and Rab4 as it is downregulated at the wavefront (not shown)". First, "not shown" is not allowed in the era of supp. data. The authors advance the hypothesis that a DI-N complex is transported by Rab7 to late endosome and that later Rab4 recycles DI to the plasma membrane. The absence of N co-localization with Rab4

supports the author's model but the fact that the authors never show a colocalization of N with Rab7 contradicts their model. They should show the data and explain why they do not see Rab7 and N just before the wavefront, before N degradation.

3. "When we knocked down Rab7 by expressing the dominant-negative form of Rab7 or Rab7 RNAi, the N protein level was slightly upregulated, showing punctate signals as visualized by the N antibody (Fig. 4a-c, l). However, the expression of the N activity reporter was not significantly affected (not shown)."

A control for Figure 4 a-c is missing.

The authors mention that N is upregulated. Here and throughout the text, the quantifications were performed by counting the number of punctate signals. This might not reflect upregulation and could simply result from a change in protein distribution resulting from the reduced N degradation.

4. "...and that the stronger RNAi effect causes the uniform non-specific N activation (Fig. 4g) while the milder effect causes the specific N activation between the twin peaks (Fig. 4d)."

In figure 4g, NRE-dVENUS is strongly downregulated, even absent in neuroepithelial cells and newborn neuroblasts. This contradicts the author's interpretation of the figure.

5. "Importantly, the specific N activation at the wavefront and the fusion of the twin peaks were reproducible in multiple RNAi conditions and mutant clones of *vps2* and *vps22* (Figs. 4h-k, S7)."

In panel K, the top clone with an arrow shows the continuous expression of the reporter of Notch activity from neuroepithelium to neuroblasts. However, the bottom clone shows an almost total absence of reporter expression. The authors should clarify these results and their interpretation.

6. "...recycling endosome, an aggregated distribution of DI protein was observed in the cytoplasm of cells at the wavefront (Fig. 5a and not shown)." All data must be shown.

How can the authors conclude that the protein is localized to the cytoplasm?

7. Lines 268-282: all results lack proper quantification.

8. Fig5g,i. It is impossible to see the results: The authors should show a higher magnification image.

Minor points

1. "However, the step function is very artificial and non-biological. According to the previous literature, which demonstrated that the kinetics of cis-inhibition is sharper than that of trans-activation".

I believe the authors want to say 'faster' and not 'sharper.'

2. "We found that the mild DI RNAi under the control of *optix-Gal4*, which is expressed in the dorsal and ventral subdomains of the optic lobe (Fig. 2d), reproduces the single N peak."

The in silico model should reproduce the biological data, not the other way around!

3. "the expression level of N should be upregulated when the function of DI is compromised. As expected, we observed striking upregulation of the full-length N protein when DI mutant clones were generated at the wavefront (Fig. 2i; n=13/25)."

Quantification of fig 2i is missing

4. "The proximity ligation assay also demonstrated that Rab7 forms a complex with 194 DI in vivo (Fig. 3d)."

It does not demonstrate, it suggests.

5. "Since the up-regulation of the level of full-length N were not as prominent as those found in DI mutant clones (Fig. 2i),"

Quantification for fig2i is missing.

6. "The Klu expression in the newborn NBs, which only express Hth in the control background, might cause the production of Run-positive neurons earlier than Bsh-positive neurons, resulting in their abnormal distributions in the medulla (Fig. 6k)."

According to the model, Bsh neurons should never be formed when Klu is precociously expressed. The authors should revisit their interpretation of the results.

7. Lines 323-329

While these are interesting observations, they do not provide any significant information to this section.

8. "When N is quickly degraded, what happens to N signaling? When N is activated by DI-mediated trans-activation, NICD is released from the plasma membrane and is translocated to the nucleus to form a transcriptional complex with Su(H). When NICD is not formed due to the degradation of N, Su(H) forms a repressor complex with H, which might trigger the quick inactivation of the N signal activity."

The authors ask a question but never answer it...

9. "It has recently been reported that cis-binding of Ser and N is inhibited by sugar modification of the extracellular domain of N by Fringe (Fng). Fng modification of N may similarly regulates the cis-interaction between DI and N. However, this idea is difficult to test because the residues that are modified by Fng are supposed to be shared between trans- and cis-interactions, and trans-activation and cis-inhibition would be simultaneously affected by Fng."

I cannot understand how this relates with the previous paragraph and what is the point the authors try to make.

10. "However, cis-inhibition may not easily fit with the oscillatory dynamics of N activity in mammalian NSCs because cis-inhibition would quickly stabilize the binary state of N activity. DI expression may be too weak to induce cis-inhibition, or the molecular mechanism of cis-inhibition may be suppressed in NPCs in the cerebral cortex. Thus, there may be multiple ways to control the temporal dynamics of N and the temporal patterning of neurogenesis."

The author's argument is difficult to follow!

11. "The neuronal type TmY14 has not been documented as far as we know." Do the authors mean that it was previously unknown that TmY14 was Runt+?

Reviewer #2 (Remarks to the Author):

Review of "Intracellular trafficking of Notch... " by the Sato lab.

Here the authors make several findings. (1) They model the spread of DI and Notch during optic lobe neurogenesis, developing a refinement of a previously published model that shows two peaks of N activity, one ahead of the NE/NB transition zone, and one behind the transition zone. High Delta is between the peaks. (2) They validate their "two peaks" of N model by in vivo N reporter analysis and DI staining. (3) They show that late endosomes are required for downregulation of N during cis-inhibition that occurs between the peaks. (4) They show that acidification of endosomes results in

release of DI from N, allowing DI to be recycled by rab4 back to the plasma membrane. (5) They show that the second peak of N is required to activate expression of Klu, the second gene in the optic lobe temporal transcription factor series. Furthermore, genotypes that collapse the two N peaks into one broad peak lead to premature expression of Klu, thereby validating their conclusion that N activates Klu in neuroblasts.

The paper is clear and logical, with good figures supporting the conclusions for points 1-4 above. There are several gaps in the temporal transcription factor analysis that should be addressed before publication.

1. In DI mutant clones (where there is no N peaks), and that lack Klu expression, what is the effect on the other temporal transcription factors? Does Hth fail to turn off? Do the subsequent factors fail to turn on? Or is Klu simply deleted and all else in the cascade is normal?
2. In DI mutant clones, what are the neuronal identity phenotypes?
3. In the DI RNAi (where there is a single N peak), what is the effect on the temporal cascade?
4. Please discuss why Klu is not activated by the first N peak.
5. In Figure 6e-h, the phenotypes are quite mild and not clearly documented. Could the authors supply more convincing images? Could the authors quantify the phenotype?

Minor comment

1. The previous work on cis-inhibition is cited (references 9-13) but the findings are not described in the Introduction. Please add a bit of text describing previous conclusions on the mechanism and function of cis-inhibition in the Introduction, or alternatively in the Discussion if you want to compare findings.

Reviewer #3 (Remarks to the Author):

In their paper, Wang and coauthors suggest a mechanism of Notch (N) cis-inhibition that explains the observation of the existence of two peaks of activity during neuroepithelial cell specification in flies. Specifically, the authors used the wave of differentiation-'proneural wave'- for revisiting their previously published mathematical model on the twin peaks. In this context, they tested whether the Delta (DI)-mediated cis-inhibition is the cause of the N downregulation after the first peak of N activity. In contrast with the previous report, a strong nonlinearity in cis-inhibition is argued to reproduce the twin peaks of N activity along the proneural wave in silico. Moreover, it is claimed that DI expression induces a quick degradation of Notch in late endosomes and the formation of the twin peaks of Notch activity in vivo.

The novelty of this paper is about three major points:

- 1) With the new mathematical arrangement, the single peak of Notch activity of the former model is replaced with a twin peaks of Notch activity that better represent the nonlinearity in cis-inhibition that occurs in vivo.
- 2) They suggest a biological mechanism that underlie the twin peaks formation. They claim that the reduction of Notch activity is due to high level of Delta at the wavefront that induce the clustering of Notch and Delta, ultimately leading to cis-inhibition via Notch degradation in late endosomes. Indeed, the downregulation of Delta or ESCRT genes leads to twin peaks fusion.
- 3) They correlate the second Notch peak to neuron differentiation through Klu transcription factor.

Although, the manuscript reports findings potentially cardinal to our understanding of proneural differentiation, the hypothesis put forward are not well supported by experimental data. In particular, in the current version, the data could be more compelling to demonstrate that late endosomes play a role in the N degradation to shape the temporal patten of neurogenesis. The authors should provide more mechanistic evidence on how cis-inhibition by DI is mediated of intracellular trafficking of N in their experimental model. Alternatively, I suggest to tone down their statement on the mechanism of N signaling inhibition.

Major comments:

- Besides genetic interactions with ESCRTs, the authors could provide mechanistic evidence on how this complex is internalized and degraded. Unclear points here are where is the N that become apparent at the plasma membrane in absence of DI or in late endosome in absence of ESCRTs or V-ATPase components? Trafficking/uptake assays at different time points would be key and can be done using anti anti- N ECD and perhaps anti-DI to follow internalization. DI-GFP could also be used. The authors could also attempt to use the latter using immuno-EM to better detail the cellular path followed by DI. If the complex reaches late endosomes in and Notch is degraded while Delta is recycled to the plasma membrane, we wouldn't expect to see DI in Rab7 compartment, even accumulating when degradation is blocked. They could also assess whether N is ever transiting in a Rab4 compartment when ESCRT are downregulated and an amount of N becomes apparent. N trafficking experiments could benefit from condition of ESCRT or DI kd or presence of fluorescent Rab proteins.
- The author claimed that Notch degradation is triggered by Delta expression at the wavefront. While this is reasonable, they should rule out that in DI mutant clones N mRNA expression is affected (eg. using Notch gene trap).
- Can overexpression of DI rescue the phenotype of vps kd?
- DI-mutated clones upregulate Notch fl. In these clones Notch activity, using for example NRE-dVenus or E(spl)my-GFP, is not shown.
- To claim that Notch is necessary to Klu expression, it is necessary to analyze what happen if Notch is downregulated (e.g. Notch KD clones should result in Klu e Run absence).
- The majority of imaging data lacks proper quantification.

Minor comments:

- The explanation of the mathematical model could be more amenable to a general public, as it is, it hard to follow
- The PLA assay should include negative controls
- The loss of Lsc is clear in Fig. 6b, less so in 6c
- There is no consistency in the image orientation. It would be clearer if all images had the same orientation
- Some graphs are without description in the caption
- Fig. 2a: Why DI antibody only marks DI clones? What about endogenous Delta (e.g. Fig. 2e) Specify the driver compartment (e.g ay-GAL4 ?).

Reviewer #4 (Remarks to the Author):

The manuscript by Wang et focuses on the role of cis-inhibition of Notch by its ligand during neurogenesis in the fly brain. More specifically, it examines the formation of twin peaks of Notch activity at the front of the pro-neural wave observed in the *Drosophila* brain. It is suggested that the two peaks are formed as a result of cis-inhibition between Notch and Delta, a process by which Delta

inhibits Notch when expressed in the same cell. Similar to an earlier model for wing veins, dynamic expression of Delta along a single stripe leads to inhibition of signaling in that stripe and activation of Notch on both sides of the stripe. Unlike the vein, this is a dynamic process that propagates with the proneural wave. Based on a mathematical model, the authors argue that cis-inhibition should lead to strong degradation of Notch in a Delta dependent manner. Focusing on the mechanism of degradation they suggest that Notch-Delta complexes are internalized via late endosomes leading to degradation of Notch and recycling of Delta back to the surface. Finally, they show that the formation of twin peaks of Notch activity is important for the proper activating sequence of different neuron types.

The manuscript presents an interesting new example for the role of cis-inhibition in-vivo and proposes a new mechanism regulating cis-inhibition between Notch and Delta. Both of these aspects would be of interest to the community. There are however some issues that needs to be resolved before the manuscript can be considered for publication:

Major issues:

1. Problems with the formulation of the model. The manuscript uses a mathematical model to quantitatively explain the twin peak profile of Notch activity (Fig. 1e-g). In the model, Notch activity (N) in a cell is activated by Delta in the neighboring cell (trans-activation) and repressed by Delta in the same cells (cis-inhibition). The authors show that the cis-inhibition term needs to have non-linearity (threshold or Hill function) in order to achieve the twin peaks. There are a few problems with the formulation of the model.

(a) The Notch activity can get negative values (see Fig. S1, S2). This is of course non-physiological. The authors argue that this could be due to the repressor Hairless, but really this does not make sense since the repressor is not defined as part of the model.

(b) The model lacks the description of what happens to full length Notch. This is particularly interesting since in Fig. 2h it is shown that full length Notch drops sharply at the edge of the 1st N activity stripe and slowly ramps up towards the back of the of the 2nd N activity stripe.

Resolving both of these issues can be achieved by introducing full length Notch into the equations (as was done in ref 11). In that case cis-inhibition can be introduced as a non-linear term that depends on the product of Notch and Delta. The sharpness of cis inhibition can be tuned by the cis-interaction parameter in that case.

Importantly, the authors should show whether the profiles of both full length Notch and Delta agree with the observed experiments (in addition to the profile of Notch activity). I am curious to see whether the model can account for the sharp boundary of full length Notch.

2. The Delta-Notch recycling model. The authors argue that N-DI complexes recycle to the late endosomes in a Rab7 dependent manner, where N degrade via V-ATPase, but DI is recycled back to the membrane via Rab4. The data however only partially supports this model:

(a) It is unclear where DI endosome are observed. It is unclear if DI endosomes are only (or mostly) observed where Notch is cis inhibited (between the tween peaks). This should be quantified by the authors. Is the localization of DI, Rab7, and Rab4 observed in this region? In general, it would be good to map where Delta is observed at the cell surface and where it is observed in endosomes, both in WT and in the mutants.

(b) The authors say that Notch puncta colocalized with Rab7 and Rab 4 are not observed, however they do not provide the images (in fact the authors should present all data quoted as "not shown"). Do they see Notch and Delta colocalize at endogenous levels? The authors should discuss why not since it does not fit their model.

(c) Notch puncta are observed when knocking down Rab7 or ESCRT components. Is DI observed to be localized with N in these puncta? If not, the authors should explain how such observation fits with their model (as I don't understand it).

(d) The only place N and DI are shown in the same image are in Fig. 5e-i. These figures refer to a very strange experiment which is hardly explained. The authors use DI-GFP line which is not described anywhere in the text. Presumably this is an overexpression line. Why is it used only here? What does

overexpression of DI-GFP does to the system? How is it interpreted? Can it be used in other experiments? This is really unclear.

In summary – more clear data and analysis are needed to establish the recycling model.

3. Activation downstream of the 2nd Notch peak. Fig. 6g-h shows that knocking down DI and vps2 leads to misorganization of the two neuron types. However, the mismatch is restricted to only one region. The authors should discuss why is the effect limited to only one region.

4. Some parts of the discussion are unclear – in particular the paragraph starting at row 350 (the point is unclear), the paragraph starting at row 394 (the point about the co-repressor seems irrelevant or at least unclear), and the last paragraph about NPC oscillation in vertebrates (is the point simply to say there are different mechanisms in different neuronal processes?)

Minor issues:

5. The authors should also discuss whether Delta activity is inhibited by cis-interaction with Notch (e.g. Becam et al 2010).

6. OPC in row 409 is not defined

REVIEWER COMMENTS

Reviewer #1 (Remarks to the Author):

In the fly optic lobe, a wave of differentiation regulates the transformation of neuroepithelial cells into dividing neuroprogenitors (neuroblasts in the fly), that over time produce distinct neuronal progeny as they transit through a transcription factor temporal series. This cell fate transition resembles the transition of neuroepithelial to radial glial cells in the developing cerebral cortex.

Several studies (including those from the senior author) have shown how this transition involves a complex regulation of Notch signaling. However, unlike what is typical for Notch signaling, the spatial activation of the pathway does not form a salt and pepper pattern. Instead, a wave of Notch activity propagates across the neuroepithelium to regulate its differentiation into neuroblasts. However, Notch has to be downregulated for the differentiation to occur and newborn neuroblasts do not show Notch activity. A second peak of Notch activity is observed as neuroblasts age. Thus, two peaks of Notch activity are under tight spatial and temporal control. The authors have previously published several exciting articles that provide a convincing model explaining how the first wave of Notch is formed and propagates across the neuroepithelium to regulate neuroblast production. However, their previous model could not explain the formation of the second wave of Notch that forms in older neuroblasts.

Here, the authors revisit their previous model. They provide evidence that this pattern of temporal Notch activation is dependent on cis-inhibition and relies on intracellular trafficking of Notch and Delta that leads to Notch degradation in late endosomes. They further show that the second wave of Notch activity in neuroblasts is required for the expression of the temporal factor Klu,

This paper provides an important contribution in the understanding of how neurogenesis and neural fate specification are orchestrated in the developing brain. However, I have some concerns regarding the data and their interpretation. In addition, the paper requires textual revisions, as it is often difficult to understand the points the authors are trying to raise. Therefore, the manuscript cannot be published before the following points are addressed.

My first recommendation is that the authors revise the text and work hard at writing clear

and concise descriptions of the experiments and of the conclusions. For example, the abstract never explains **what the “twin peaks” are**, which makes the question confusing for the reader.

We would like to thank you for raising this critical point. We added a description that there is a second peak of Notch activity in the abstract as follows (Line 21-23). Since there is a first and second peaks, the meaning of the twin peaks will become easier to understand:

we show that strong nonlinearity in *cis*-inhibition reproduces the second peak of Notch activity behind the proneural wave *in silico*.

Furthermore, although the authors mention that they have a prior mathematical model explaining the kinetics of the proneural wave, **they never mention that this model utilizes Achaete-Scute and EGF signaling to model the wave, and the figure legends/text never mention what A and E stand for (leaving the reader to refer to the methods to decipher things)**. In addition to these points, other points might have been missed because of the unclear writing (see below).

Thank you for the constructive comment. We added an explanation about the previous mathematical model in Introduction (Line 57-62):

In the previous study, we formulated a mathematical model of the proneural wave, which includes N activity (*N*), D1 expression (*D*), EGF signal activity (*E*) and the state of NB differentiation (*A*). *A* is related to the expression levels of Achaete-Scute Complex proteins (AS-C). The model successfully reproduces the complex behaviors of the proneural wave in various genetic backgrounds^{25,26}.

Major points

1. For all manipulations in which the two waves of Notch activity are lost, the authors' interpretation is that the two waves are fused. A close look at their model (Figure 1g) shows this is not the case. Strong D1 expression creates a decrease in Notch activity which is at the origin of the two waves. A reduction of its expression leads to reduced *cis*-inhibition: As a consequence, Notch is never downregulated and its expression is maintained from NE to D1 expressing cells to aging neuroblasts.

Thank you for giving us a critical comment. We made the following explanations why we use the word “fused” to describe the merge of the two peaks into one broad peak.

In physics, we can say 'soliton fusion'. Since soliton is a type of wave, it is not necessarily made from physical objects. When two solitons fuse with each other, the signal or activity between them increase filling the gap between them, just like the

fusion of the twin peaks of Notch activity. Therefore, the use of 'fused' for the two peaks Notch activity should not be a problem, if it is allowed in physics.

Since we are not looking at objects like cells, the twin peaks of Notch activity are not physically fusing with each other. When Notch activity between the two activity peaks increase filling the gap between them, they behave as if two objects are fusing. Although the phenomenon we observe is not a fusion of the objects, the two activity peaks behave as if they are fusing just like soliton. We therefore chose the word 'fused' to describe this phenomenon.

2. "We could not detect co-localization of N with Rab7 and Rab4 as it is downregulated at the wavefront (not shown)". First, "not shown" is not allowed in the era of supp. data. The authors advance the hypothesis that a DI-N complex is transported by Rab7 to late endosome and that later Rab4 recycles DI to the plasma membrane. The absence of N co-localization with Rab4 supports the author's model but the fact that the authors never show a colocalization of N with Rab7 contradicts their model. They should show the data and explain why they do not see Rab7 and N just before the wavefront, before N degradation.

We would like to thank you for pointing this question out. Although N is down regulated at the wave front, we occasionally observed minor colocalization of N with Rab7 and Rab4 in the wild type brain (Fig. 3j). Similarly, co-localization of N and DI-GFP was occasionally observed (Figs. 3l). The co-localizations of N with Rab7/4 were less significant compared with those of DI with Rab7/4 (Fig. 3k). Importantly, N co-localized with Rab7 more significantly compared with Rab4, supporting the hypothesis that N is mainly degraded in late endosomes at the wave front in a DI-dependent manner (Line 216-224).

Although N is down regulated at the wave front, we occasionally observed minor colocalizations of N with Rab7 and Rab4 (Fig. 3j). The co-localization indices of N and Rab7/4 were significantly lower compared with those of DI and Rab7/4 (Fig. 3k). Importantly, N co-localized with Rab7 more significantly compared with Rab4, suggesting that Rab7-dependent N-degradation is more dominant than Rab4-dependent N recycling. Similarly, co-localization of N and DI-GFP, which recapitulates DI distribution pattern, was occasionally observed (Fig. 3l, Supplementary Figure 5c). These results support the hypothesis that N is mainly degraded in late endosomes at the wave front in a DI-dependent manner.

We also performed additional experiments to see their co-localizations when ESCRT complex genes are knocked down. When *vps2* or *vps25* was knocked down, N was

aggregated at the wavefront and colocalized with DI in Rab7 positive puncta (Fig. 4e-h, j). Furthermore, N was aggregated and co-localized with Rab4 puncta (Fig. 4i). We also demonstrated that N colocalizes with Rab4 when acidification of late endosomes is compromised in *vha68-2* RNAi backgrounds (Fig. 6j).

3. “When we knocked down Rab7 by expressing the dominant-negative form of Rab7 or Rab7 RNAi, the N protein level was slightly upregulated, showing punctate signals as visualized by the N antibody (Fig. 4a-c, l). However, the expression of the N activity reporter was not significantly affected (not shown).’ A control for Figure 4 a-c is missing. The authors mention that N is upregulated. Here and throughout the text, the quantifications were performed by counting the number of punctate signals. This might not reflect upregulation and could simply result from a change in protein distribution resulting from the reduced N degradation.

As indicated by Reviewer #1, we added controls for these figures (Fig. 4a, c) and quantified the results by measuring the intensity of N expression in control and RNAi area (Figs. 4j, 5n).

4. “.....and that the stronger RNAi effect causes the uniform non-specific N activation (Fig. 4g) while the milder effect causes the specific N activation between the twin peaks (Fig. 4d).”

In figure 4g, NRE-dVENUS is strongly downregulated, even absent in neuroepithelial cells and newborn neuroblasts. This contradicts the author’s interpretation of the figure. We apologize for our confusing explanation. In Fig. 5m (previous Fig. 4g), NRE-dVenus signals were not eliminated, but rather uniformly activated at a low level. It has been shown that *vps2* plays multiple roles in controlling N activity either positively or negatively. We modified the description of the phenotype as follows (line 266-273): In the other 60% of the cases, DI and full-length N proteins were widely upregulated (Figs. 4h, j, 5m, Supplementary Figure 6c, d; n=12/20). Low level N activity was observed in a wide area encompassing the wavefront (Fig. 5m, Supplementary Figure 6c), which may be related to the hyper activation of N signaling and/or suppression of N activity in *vps2* mutant cells^{15-18,20}. Since these phenotypes accompany smaller brain size compared with the brains showing the specific N activation phenotype discussed above, we assume that the mild RNAi effect caused the specific N activation between the twin peaks (Fig. 5d).

5. “Importantly, the specific N activation at the wavefront and the fusion of the twin

peaks were reproducible in multiple RNAi conditions and mutant clones of *vps2* and *vps22* (Figs. 4h-k, S7).”

In panel K, the top clone with an arrow shows the continuous expression of the reporter of Notch activity from neuroepithelium to neuroblasts. However, the bottom clone shows an almost total absence of reporter expression. The authors should clarify these results and their interpretation.

Thank you for raising this point. As suggested, we quantified the frequency of the phenotype. Upregulation of N activity was frequently found in *vps2* and *vps22* mutant clones (n=9/15 and 11/16, Fig. 5o, p), but not always (line 278-280).

Note that the twin peaks were partially fused while ectopic N activation was not found in cells apart from the wavefront in the *vps2* and *vps22* mutant clones (Fig. 5o, p; n=9/15 and 11/16).

6. “...recycling endosome, an aggregated distribution of DI protein was observed in the cytoplasm of cells at the wavefront (Fig. 5a and not shown).” All data must be shown. How can the authors conclude that the protein is localized to the cytoplasm?

As indicated by Reviewer, we showed all the data about DI distribution upon *rab4* knock down in Fig. 6a-c, Supplementary Figure 9 (Line 291-300). We observed that DI mainly aggregated in the cytoplasm by co-staining with TO-PRO3 and Ecad (Supplementary Figure 9).

When we knocked down Rab4 with RNAi, DI expression was accumulated at the wave front in a milder condition at 25°C (Fig. 6a, b, m) and the colocalization of DI with Rab7 was significantly increased (Fig. 6b). Interestingly, DI expression was downregulated in a stronger RNAi condition at 30°C (Fig. 6c, m). These results suggest that DI is retained in Rab7-positive late endosomes and is degraded together with N when the function of recycling endosomes is eliminated. Furthermore, overexpression of Rab4^{DN} mimicked the effects of *rab4* RNAi in the milder condition, and aggregated distribution of DI colocalized with Rab7 in the cytoplasm of cells at the wavefront (Supplementary Figure 9).

7. Lines 268-282: all results lack proper quantification.

As suggested, we added quantifications in Fig. 6b, c, m, n, Supplementary Figure 10c.

8. Fig5g, i. It is impossible to see the results: The authors should show a higher magnification image.

As suggested, we added higher magnification images in Fig. 6g-l.

Minor points

1. “However, the step function is very artificial and non-biological. According to the previous literature, which demonstrated that the kinetics of cis-inhibition is sharper than that of trans-activation”.

I believe the authors want to say ‘faster’ and not ‘sharper.’

As suggested, we replaced the word ‘sharper’ with ‘faster’ (Line 130, 139).

2. “We found that the mild DI RNAi under the control of optix-Gal4, which is expressed in the dorsal and ventral subdomains of the optic lobe (Fig. 2d), reproduces the single N peak.”

The in silico model should reproduce the biological data, not the other way around!

As suggested, we replaced 'reproduces' with 'causes' (Line 149).

3. “the expression level of N should be upregulated when the function of DI is compromised. As expected, we observed striking upregulation of the full-length N protein when DI mutant clones were generated at the wavefront (Fig. 2i; n=13/25).”

Quantification of fig 2i is missing

We added a quantification plot in Fig. 2i.

4. “The proximity ligation assay also demonstrated that Rab7 forms a complex with DI in vivo (Fig. 3d).”

It does not demonstrate, it suggests.

As suggested, we replaced 'demonstrate' with 'suggest' (Line 204).

5. “Since the up-regulation of the level of full-length N were not as prominent as those found in DI mutant clones (Fig. 2i),”

Quantification for fig2i is missing.

We added a quantification plot in Fig. 2i.

6. “The Klu expression in the newborn NBs, which only express Hth in the control background, might cause the production of Run-positive neurons earlier than Bsh-positive neurons, resulting in their abnormal distributions in the medulla (Fig. 6k).”
According to the model, Bsh neurons should never be formed when Klu is precociously expressed. The authors should revisit their interpretation of the results.

Hth is widely expressed both in NE and NB cells and not affected by Klu (Suzuki et al., 2013). Thus, production of Bsh-positive neurons should not be inhibited by Klu. The following sentences were added (Line 370-372).

Note that Hth expression is widely found in NE and NB cells, and is not affected by Klu³¹. Therefore, expression of Hth and production of Bsh-positive neurons should not be affected.

7. Lines 323-329

While these are interesting observations, they do not provide any significant information to this section.

We use Run as a marker for neuron subtypes. Since neuronal subtypes are classified according to the projection patterns of neurons in the optic lobe, it is essential to show the projection patterns of Run-positive neurons. We modified the sentence as follows (Line 376-377).

The neuronal type TmY14 has not been documented as far as we know based on its projection pattern in the medulla, lobula and lobula plate.

8. “When N is quickly degraded, what happens to N signaling? When N is activated by DI-mediated trans-activation, NICD is released from the plasma membrane and is translocated to the nucleus to form a transcriptional complex with Su(H). When NICD is not formed due to the degradation of N, Su(H) forms a repressor complex with H, which might trigger the quick inactivation of the N signal activity.”

The authors ask a question but never answer it....

We removed this paragraph since we did not directly test the behaviors of Su(H) and H in this study.

9. “It has recently been reported that cis-binding of Ser and N is inhibited by sugar modification of the extracellular domain of N by Fringe (Fng). Fng modification of N may similarly regulates the cis-interaction between DI and N. However, this idea is difficult to test because the residues that are modified by Fng are supposed to be shared between trans- and cis-interactions, and trans-activation and cis-inhibition would be simultaneously affected by Fng.”

I cannot understand how this relates with the previous paragraph and what is the point the authors try to make.

We removed this paragraph since these observations are not directly related to the current study.

10. "However, cis-inhibition may not easily fit with the oscillatory dynamics of N activity in mammalian NSCs because cis-inhibition would quickly stabilize the binary state of N activity. Dl expression may be too weak to induce cis-inhibition, or the molecular mechanism of cis-inhibition may be suppressed in NPCs in the cerebral cortex. Thus, there may be multiple ways to control the temporal dynamics of N and the temporal patterning of neurogenesis."

The author's argument is difficult to follow!

Since the roles of cis-inhibition in mammalian NSC differentiation have not been studied so far, we changed this paragraph as follows (Line 432-438):

In the neural progenitor cells (NPCs) of the developing cerebral cortex, the temporal dynamics of N activity also plays important roles in the temporal patterning of neurogenesis and gliogenesis⁵³. In this process, the basic helix-loop-helix transcription factors show oscillatory expression in NPCs and N signaling appears to perform lateral inhibitory feedback during NPC differentiation. The roles of *cis*-inhibition in this process remain largely elusive. It will be interesting to see how the molecular mechanisms revealed in the current study are conserved in a wide variety of developmental processes.

11. "The neuronal type TmY14 has not been documented as far as we know." Do the authors mean that it was previously unknown that TmY14 was Runt+?

TmY14 neuron itself was previously unknown (Line 376-377).

The neuronal type TmY14 has not been documented as far as we know based on its projection pattern in the medulla, lobula and lobula plate.

Reviewer #2 (Remarks to the Author):

Review of "Intracellular trafficking of Notch... " by the Sato lab.

Here the authors make several findings. (1) They model the spread of Dl and Notch during optic lobe neurogenesis, developing a refinement of a previously published model that shows two peaks of N activity, one ahead of the NE/NB transition zone, and one behind the transition zone. High Delta is between the peaks. (2) They validate their "two peaks" of N model by in vivo N reporter analysis and Dl staining. (3) They show that late endosomes are required for downregulation of N during cis-inhibition that occurs between the peaks. (4) They show that acidification of endosomes results in release of Dl from N, allowing Dl to be recycled by rab4 back to the plasma membrane. (5) They show that the second peak of N is required to activate expression of Klu, the second gene in the

optic lobe temporal transcription factor series. Furthermore, genotypes that collapse the two N peaks into one broad peak lead to premature expression of Klu, thereby validating their conclusion that N activates Klu in neuroblasts.

The paper is clear and logical, with good figures supporting the conclusions for points 1-4 above. There are several gaps in the temporal transcription factor analysis that should be addressed before publication.

1. In *Dl* mutant clones (where there is no N peaks), and that lack Klu expression, what is the effect on the other temporal transcription factors? Does Hth fail to turn off? Do the subsequent factors fail to turn on? Or is Klu simply deleted and all else in the cascade is normal?

We would like to thank the reviewer for this question. We performed additional experiments to examine the expression of the other transcription factors in *Dl* mutant clones and *Dl* RNAi backgrounds. The results showed that expression of Hth, Ey and Slp were not affected (Supplementary Figure 11, Line 348-349), suggesting that N activity specifically regulates Klu expression without affecting the other temporal transcription factors. Furthermore, the loss of Klu expression does not affect the temporal expression of the following transcription factors (Suzuki et al, Dev Biol 380, 12-24, 2013).

In contrast, the expression Hth, Ey and Slp was not significantly affected in *Dl* mutant clones (Supplementary Figure 11a-c).

2. In *Dl* mutant clones, what are the neuronal identity phenotypes?

The outcome of Notch activity in *Dl* mutant clones depends on the expression of *Dl* in neighboring cells. Notch signaling should be activated in the *Dl* mutant cells that are adjacent to *Dl* expressing cells, while inactivated when they are isolated from *Dl* expression. Therefore, the phenotypes of *Dl* mutant NBs should be diverse. Additionally, those NBs produce many GMCs and neurons. It has been reported that Notch signaling is involved in the asymmetric cell type specification in GMCs, which might also be influenced by *Dl* (Li et al., 2013). If we generate *Dl* mutant clones in NEs and NBs, the clones produce many cells with very diverse neuronal identity phenotypes. Though it is an interesting question, we did not try this experiment because the results would be very confusing.

3. In the DI RNAi (where there is a single N peak), what is the effect on the temporal cascade?

We did additional experiments to examine the expression of the other transcription factors in DI mutant clones and DI RNAi backgrounds. Expression of Hth, Ey and Slp was not affected (Supplementary Figure 11, Line 348-349).

4. Please discuss why Klu is not activated by the first N peak.

Thank you for the helpful comment. NEs and NBs are the different cell types. Klu expression may require additional genetic factors that are specific to NBs. We added the following sentences (Line 356-358).

Note that Klu expression is not activated in the first peak of N activity in NEs (Fig. 7a, l). Klu expression may require additional genetic factors that are specific to NBs.

5. In Figure 6e-h, the phenotypes are quite mild and not clearly documented. Could the authors supply more convincing images? Could the authors quantify the phenotype?

We apologize for the confusing presentation of the results. optix-Gal4 is expressed in the dorsal and ventral parts of the brain showing stronger signals in the dorsal part (Fig. 7e). When RNAi was induced under the control of optix-Gal4, the stronger phenotype was usually observed in the dorsal part of the brain. We quantified the phenotypes by measuring the positions of the cells. We quantified the size of the overlap between Run- and Bsh-positive areas. The overlap was significantly increased in the dorsal RNAi area compared with the control areas (Fig. 7g-i).

Minor comment

1. The previous work on cis-inhibition is cited (references 9-13) but the findings are not described in the Introduction. Please add a bit of text describing previous conclusions on the mechanism and function of cis-inhibition in the Introduction, or alternatively in the Discussion if you want to compare findings.

As suggested, previous works on the mechanisms of cis-inhibition are described in Line 42-54.

On the other hand, N is autonomously inactivated by DI expressed in the same cell through a process 'cis-inhibition', whose molecular mechanism and biological significance remain largely elusive⁹⁻¹².

The direct interaction between DI and N seems to trigger *cis*-inhibition by inhibiting N prior to or following its transport to the plasma membrane¹³. There are two possible mechanisms of *cis*-inhibition. First, the *cis*-interaction of the ligand and receptor

may shut off the transport of N from the ER to the plasma membrane¹⁴. Second, the *cis*-interaction may trigger the catalytic process that results in N degradation. For example, the DI-N complex may be internalized from the plasma membrane to cause N degradation. Protein degradation in late endosomes has been shown to play important roles in activating and inactivating N signaling during *trans*-activation¹⁵⁻²⁴. However, the potential roles of intracellular trafficking of DI and N in *cis*-inhibition remain largely unknown.

Reviewer #3 (Remarks to the Author):

In their paper, Wang and coauthors suggest a mechanism of Notch (N) *cis*-inhibition that explains the observation of the existence of two peaks of activity during neuroepithelial cell specification in flies. Specifically, the authors used the wave of differentiation-'proneural wave'- for revisiting their previously published mathematical model on the twin peaks. In this context, they tested whether the Delta (DI)-mediated *cis*-inhibition is the cause of the N downregulation after the first peak of N activity. In contrast with the previous report, a strong nonlinearity in *cis*-inhibition is argued to reproduce the twin peaks of N activity along the proneural wave *in silico*. Moreover, it is claimed that DI expression induces a quick degradation of Notch in late endosomes and the formation of the twin peaks of Notch activity *in vivo*.

The novelty of this paper is about three major point:

- 1) With the new mathematical arrangement, the single peak of Notch activity of the former model is replaced with a twin peaks of Notch activity that better represent the nonlinearity in *cis*-inhibition that occurs *in vivo*.
- 2) They suggest a biological mechanism that underlie the twin peaks formation. They claim that the reduction of Notch activity is due to high level of Delta at the wavefront that induce the clustering of Notch and Delta, ultimately leading to *cis*-inhibition via Notch degradation in late endosomes. Indeed, the downregulation of Delta or ESCRT genes leads to twin peaks fusion.
- 3) They correlate the second Notch peak to neuron differentiation through Klu transcription factor.

Although, the manuscript reports findings potentially cardinal to our understanding of proneural differentiation, the hypothesis put forward are not well supported by experimental data. In particular, in the current version, the data could be more compelling to demonstrate that late endosomes play a role in the N degradation to shape the temporal patten of neurogenesis. **The authors should provide more mechanistic evidence on how *cis*-inhibition by DI is mediated of intracellular trafficking of N in their**

experimental model. Alternatively, I suggest to tone down their statement on the mechanism of N signaling inhibition.

We thank the reviewer for critical comments. Although the uptake assay and immuno-EM analyses are technically very difficult in our system focusing on the fly brain in vivo, we extended our immunohistochemical and genetic approaches to address the trafficking of N and DI as follows (see below). Additionally, we emphasized that our idea is a hypothesis, which is supported by our data.

Major comments:

- Besides genetic interactions with ESCRTs, the authors could provide mechanistic evidence on how this complex is internalized and degraded. Unclear points here are where is the N that become apparent at the plasma membrane in absence of DI or in late endosome in absence of ESCRTs or V-ATPase components?

Thank you for raising this important point. We examined the localization of N in the absence of DI by co-staining with DE-cadherin. In DI mutant clones, N was accumulated along the plasma membrane visualized by E-cadherin staining (Fig. 2j).

We also demonstrated that N and DI are co-localized with Rab7 and Rab4 when ECSRT function is compromised in *vps2* or *vps25* RNAi (Fig. 4e-j). Similar results were obtained when V-ATPase function was compromised in *vha68-2* RNAi (Fig. 6g-j). These results suggest that DI-N complex stays in late endosomes or is transported to recycling endosomes when N is not degraded.

Trafficking/uptake assays at different time points would be key and can be done using anti anti- N ECD and perhaps anti-DI to follow internalization. DI-GFP could also be used. The authors could also attempt to use the latter using immuno-EM to better detail the cellular path followed by DI.

We agree that the suggested experiments would provide a direct mechanistic view on the intercellular trafficking of DI and N in a cell culture system. However, we focus on the proneural wave progression in the fly brain in vivo in this study. Since the brain is wrapped with the surface glial layers that act as a tight diffusion barrier, DI and N expressed in NEs and NBs are not directly accessible in the culture medium. We need to rely on the genetic and immunohistochemical approaches to deal with the fly brain in vivo. Establishing a new experimental system for the trafficking/uptake assays in the cultured brain combined with immuno-EM sounds exciting, but are beyond the scope of this study. Instead, we extended our immunohistochemical and genetic approaches to address the trafficking of N and DI as follows (Figs. 3, 4, 6).

If the complex reaches late endosomes in and Notch is degraded while Delta is recycled to the plasma membrane, we wouldn't expect to see DI in Rab7 compartment, even accumulating when degradation is blocked.

Thank you for giving us critical comments. We added the following experiments to address the concerns:

1. Rab4 signals are found inside the DI puncta, while Rab7 mainly accumulate on the surface of the DI puncta at a higher magnification (Fig. 3i). The co-localization index of DI-Rab4 was greater compared with that of DI-Rab7. The less prominent colocalization of DI with Rab7 may explain why DI is not degraded in late endosomes but are mainly recycled to the plasma membrane in the control backgrounds.
2. Although N is down regulated at the wave front in the wild type brain, we occasionally observed minor colocalizations of N with Rab7 and Rab4 (Fig. 3j). Similarly, co-localization of N and DI-GFP was occasionally observed (Figs. 3l). The co-localization of N and Rab7/4 were less significant compared with those of DI and Rab7/4 (Fig. 3k). Importantly, N co-localized with Rab7 more significantly compared with Rab4, supporting the hypothesis that N is mainly degraded in late endosomes at the wave front.
3. DI expression was tested in different *rab4* RNAi conditions. In a milder condition at 25°C, DI showed aggregated distribution strongly colocalizing with Rab7, suggesting that partial reduction of DI recycling may increase DI accumulation in late endosomes (Fig. 6a, b, m). In a stronger condition at 30°C, DI expression was significantly reduced (Fig. 6c, m). DI may be degraded in late endosomes, if it's recycling to the plasma membrane is strongly inhibited.
4. pH increase upon *vha68-2* RNAi caused DI colocalization with Rab7 and Rab4 (Fig. 6h, i), suggesting that DI-N complex may be localized to either late endosomes or recycling endosomes when acidification is compromised. According to the hypothesis, DI should more strongly colocalize with Rab7 when Rab4 is knocked down together with *vha68-2*. The colocalization between DI and Rab7 was significantly increased by knocking down *rab4* together with *vha68-2* (Fig. 6h, k, l, n).

These results support our hypothesis that DI-N complex is transported to late endosomes, where only N is degraded, and that DI is released from late endosomes and is recycled to the plasma membrane through recycling endosomes (Fig. 6o).

They could also assess whether N is ever transiting in a Rab4 compartment when ESCRT are downregulated and an amount of N becomes apparent. N trafficking experiments could benefit from condition of ESCRT or DI kd or presence of fluorescent Rab proteins. As suggested by the reviewer, we demonstrated that N is upregulated and co-localized with Rab4 when ESCRT function is compromised in *vps25* RNAi (Fig. 4i, j). We also demonstrated that N is accumulated and co-localized with Rab4 when V-ATPase (*vha68-2*) function is compromised (Fig. 6j).

- The author claimed that Notch degradation is triggered by Delta expression at the wavefront. While this is reasonable, they should rule out that in DI mutant clones N mRNA expression is affected (eg. using Notch gene trap).

Thank you for the constructive comment. We confirmed that N mRNA distribution was uniform at the wavefront and was not affected in *DI* mutant clones by *in situ* hybridization (Supplementary Figure 5b). These results suggest that post-translational N degradation upon DI expression is the basis of the nonlinear nature of cis-inhibition.

- Can overexpression of DI rescue the phenotype of *vps* kd?

Overexpression of DI autonomously induces cis-inhibition, and non-autonomously activates N signaling as shown in Fig. 2a. Therefore, the loss of cis-inhibition caused by *vps* RNAi would not be simply rescued by DI overexpression.

- DI-mutated clones upregulate Notch fl. In these clones Notch activity, using for example NRE-dVenus or E(spl) μ -GFP, is not shown.

As suggested, we examined Notch activity in DI mutant clones by using NRE-dVenus (Supplementary Figure 4c). As expected, N signaling is essentially inactivated in DI mutant clones because DI ligand is required for trans-activation even when full-length N is upregulated.

- To claim that Notch is necessary to Klu expression, it is necessary to analyze what happen if Notch is downregulated (e.g. Notch KD clones should result in Klu e Run absence).

Thank you for pointing this question out. We added a data showing that Klu expression was eliminated in N mutant clones (Supplementary Figure 11f).

- The majority of imaging data lacks proper quantification.

According to the suggestion, we added quantifications in Figs. 2i, 3i, 3k, 4j, 5b, 5c, 5h, 5i, 5k, 5l, 5n, 6b, 6c, 6m, 6n, 7i, Supplementary Figure 10c.

Minor comments:

- The explanation of the mathematical model could be more amenable to a general public, as it is, it hard to follow

We added explanations of the previous mathematical model in line 57-62. In the first Results section, we added explanations of step function and Hill function without using technical terms (line 117-120, 125-127).

- The PLA assay should include negative controls

We added a negative control by using Rab11 antibody (Fig. 3g).

- The loss of Lsc is clear in Fig. 6b, less so in 6c.

We apologize for the confusing data presentation. As the proneural wave is accelerated, the wavefront expressing Lsc becomes irregular. Incorporating the out-of-focus Lsc signals, the improved images in Fig. 7b, c show that the wavefront cells express Lsc.

- There is no consistency in the image orientation. It would be clearer if all images had the same orientation

As suggested, we adjusted the orientations of the images (Fig. 7e-h).

- Some graphs are without description in the caption

We added description about the graphs in Legends.

- Fig. 2a: Why D1 antibody only marks D1 clones? What about endogenous Delta (e.g. Fig. 2e) Specify the driver compartment (e.g. *ay-GAL4* ?).

The endogenous Delta expression is weaker than the overexpressed Delta expression. To see the endogenous Delta expression, the brightness of the image was enhanced to show the endogenous D1 expression as indicated by the yellow arrow (Fig. 2a). *AyGal4* is described in Methods and Legends (Line 511, 515, 799).

Reviewer #4 (Remarks to the Author):

The manuscript by Wang et focuses on the role of cis-inhibition of Notch by its ligand during neurogenesis in the fly brain. More specifically, it examines the formation of twin peaks of Notch activity at the front of the pro-neural wave observed in the *Drosophila*

brain. It is suggested that the two peaks are formed as a result of cis-inhibition between Notch and Delta, a process by which Delta inhibits Notch when expressed in the same cell. Similar to an earlier model for wing veins, dynamic expression of Delta along a single stripe leads to inhibition of signaling in that stripe and activation of Notch on both sides of the stripe. Unlike the vein, this is a dynamic process that propagates with the proneural wave. Based on a mathematical model, the authors argue that cis-inhibition should lead to strong degradation of Notch in a Delta dependent manner. Focusing on the mechanism of degradation they suggest that Notch-Delta complexes are internalized via late endosomes leading to degradation of Notch and recycling of Delta back to the surface. Finally, they show that the formation of twin peaks of Notch activity is important for the proper activating sequence of different neuron types.

The manuscript presents an interesting new example for the role of cis-inhibition in-vivo and proposes a new mechanism regulating cis-inhibition between Notch and Delta. Both of these aspects would be of interest to the community. There are however some issues that needs to be resolved before the manuscript can be considered for publication:

Major issues:

1. Problems with the formulation of the model. The manuscript uses a mathematical model to quantitatively explain the twin peak profile of Notch activity (Fig. 1e-g). In the model, Notch activity (N) in a cell is activated by Delta in the neighboring cell (trans-activation) and repressed by Delta in the same cells (cis-inhibition). The authors show that the cis-inhibition term needs to have non-linearity (threshold or Hill function) in order to achieve the twin peaks. There are a few problems with the formulation of the model.

(a) The Notch activity can get negative values (see Fig. S1, S2). This is of course non-physiological. The authors argue that this could be due to the repressor Hairless, but really this does not make sense since the repressor is not defined as part of the model.

We would like to thank the reviewer for raising this important point. The negative values of N activity might be explained by the repressor function of Hairless. However, we do not have any direct evidence that supports this interpretation. So, we only used the parameter settings with which the variable N remains non-negative in this study as added below (Line 185-186).

We only use the parameter settings with which N remains non-negative in the following study.

(b) The model lacks the description of what happens to full length Notch. This is particularly interesting since in Fig. 2h it is shown that full length Notch drops sharply at

the edge of the 1st N activity stripe and slowly ramps up towards the back of the of the 2nd N activity stripe. Resolving both of these issues can be achieved by introducing full length Notch into the equations (as was done in ref 11). In that case cis-inhibition can be introduced as a non-linear term that depends on the product of Notch and Delta. **The sharpness of cis inhibition can be tuned by the cis-interaction parameter in that case.** Importantly, the authors should show whether the profiles of both full length Notch and Delta agree with the observed experiments (in addition to the profile of Notch activity). I am curious to see whether the model can account for the sharp boundary of full length Notch.

We thank the reviewer for this important suggestion. According to the suggestion, we added a new mathematical model that incorporates the full-length N (F) and N signal activity (S). As shown in Supplementary Figure 12, the new model reproduces the twin peaks of N activity and the profiles of full-length Notch, which sharply decreases behind the wavefront (line 417-424).

The mathematical models in Fig. 1 do not explicitly consider the degradation of N protein upon its *cis*-interaction with DI. We further improved the model by considering full length N (F) and active form of N (S). In a wide range of parameter settings, this model reproduces the formation of the twin peaks of N activity, **fast degradation and gradual recovery of the expression level of full-length N** (Supplementary Figure 12). Although DI function is thought to be inhibited when DI and N interact in *cis*¹¹, we did not include this reaction in the model, because we do not have any observation that suggests *cis*-inhibition of DI in the course of the proneural wave progression.

2. The Delta-Notch recycling model. The authors argue that N-DI complexes recycle to the late endosomes in a Rab7 dependent manner, where N degrade via V-ATPase, but DI is recycled back to the membrane via Rab4. The data however only partially supports this model:

As indicated by reviewer, we added extra data that support the Delta-Notch recycling model as follows.

1. Rab4 signals are found inside the DI puncta, while Rab7 mainly accumulate on the surface of the DI puncta at a higher magnification (Fig. 3i). The co-localization index of DI-Rab4 was greater compared with that of DI-Rab7 (Line 210-215). The less prominent colocalization of DI with Rab7 may explain why DI is not degraded in late endosomes but are mainly recycled to the plasma membrane in the control backgrounds.

2. Although N is down regulated at the wave front in the wild type brain, we occasionally observed minor colocalizations of N with Rab7 and Rab4 (Fig. 3j). Similarly,

co-localization of N and DI-GFP was occasionally observed (Figs. 3l). The co-localizations of N and Rab7/4 were less significant compared with those of DI and Rab7/4 (Fig. 3k). Importantly, N co-localized with Rab7 more significantly compared with Rab4, supporting the hypothesis that N is mainly degraded in late endosomes at the wave front.

3. DI expression was tested in different *rab4* RNAi conditions. In a milder condition at 25°C, DI showed aggregated distribution strongly colocalizing with Rab7, suggesting that partial reduction of DI recycling may increase DI accumulation in late endosomes (Fig. 6a, b, m). Overexpression of Rab4^{DN} showed a similar result (Supplementary Figure 9). In a stronger condition at 30°C, DI expression was significantly reduced (Fig. 6c, m). DI may be degraded in late endosomes, if its recycling to the plasma membrane is strongly inhibited (Line 291-300).

4. pH increase upon *vha68-2* RNAi caused DI colocalization with Rab7 and Rab4 (Fig. 6h, i), suggesting that DI-N complex may be localized to either late endosomes or recycling endosomes when acidification is compromised. According to the hypothesis, DI should more strongly colocalize with Rab7 when Rab4 is knocked down together with *vha68-2*. The colocalization between DI and Rab7 was significantly increased by knocking down *rab4* together with *vha68-2* (Fig. 6h, k, l, n; Line 323-327).

These results support our hypothesis that DI-N complex is transported to late endosomes, where only N is degraded, and that DI is released from late endosomes and is recycled to the plasma membrane through recycling endosomes (Fig. 6o).

(a) It is unclear where DI endosome are observed. It is unclear if DI endosomes are only (or mostly) observed where Notch is cis inhibited (between the two peaks). This should be quantified by the authors. Is the localization of DI, Rab7, and Rab4 observed in this region? In general, it would be good to map where Delta is observed at the cell surface and where it is observed in endosomes, both in WT and in the mutants.

DI is colocalized with Rab7 and Rab4 only nearby the wavefront as visualized by Lsc (Fig. 3a, b). Lsc expressing region roughly coincides with DI expression peak and N down regulation (Figs. 1c, 2h), suggesting that the DI puncta are found nearby the DI expression peak and N down regulation. The triple labeling of DI-GFP, N and Rab7 shows the location of DI puncta along the wavefront (Fig. 3l).

We also observed ectopic DI/Rab7 puncta that are distant from the wavefront in *vps2* and *vha68-2* RNAi (Figs. 4h, 6h). A low level of DI apart from the wavefront might be cleared by late endosome-dependent degradation in the control backgrounds. Since the

mechanism of DI degradation is beyond the scope of this study, we did not quantify the DI puncta distant from the wavefront in these mutant backgrounds.

(b) The authors say that Notch puncta colocalized with Rab7 and Rab 4 are not observed, however they do not provide the images (in fact the authors should present all data quoted as “not shown”). Do they see Notch and Delta colocalize at endogenous levels? The authors should discuss why not since it does not fit their model.

Thank you for raising this important point. Although N is down regulated at the wave front in the wild type brain, we occasionally observed minor colocalization of N with Rab7 and Rab4 (Fig. 3j). Similarly, co-localization of N and DI-GFP was occasionally observed (Fig. 3l). The co-localizations of N and Rab7/4 were less significant compared with those of DI and Rab7/4 (Fig. 3k). Importantly, N co-localized with Rab7 more significantly compared with Rab4, supporting the hypothesis that N is mainly degraded in late endosomes at the wave front. Additionally, we observed N puncta that colocalizes with DI and Rab7 in *vps2* and *vha68-2* RNAi backgrounds (Figs. 4e, f, g, 6g-j).

(c) Notch puncta are observed when knocking down Rab7 or ESCRT components. Is DI observed to be localized with N in these puncta? If not, the authors should explain how such observation fits with their model (as I don't understand it).

As indicate by the reviewer, we detected the co-localization between N, DI and Rab7 in *vps2* RNAi when the ESCRT components function was blocked (Fig. 4e, f, g). We observed upregulation of N that co-localized with DI and Rab7 in many puncta.

(d) The only place N and DI are shown in the same image are in Fig. 5e-i. These figures refer to a very strange experiment which is hardly explained. The authors use DI-GFP line which is not described anywhere in the text. Presumably this is an overexpression line. Why is it used only here? What does overexpression of DI-GFP does to the system? How is it interpreted? Can it be used in other experiments? This is really unclear.

In summary – more clear data and analysis are needed to establish the recycling model. We are sorry for the confusing description. DI-GFP is a protein trap line described in the Methods section (Line 503). And this line is NOT an overexpression line. We need to use DI-GFP when we compare the distributions of N and DI, because we cannot perform double staining for mouse anti-N and mouse anti-DI antibodies. We confirmed that signals of DI-GFP and DI antibody overlap with each other (Supplementary Figure 5c).

3. Activation downstream of the 2nd Notch peak. Fig. 6g-h shows that knocking down DI

and *vps2* leads to mis-organization of the two neuron types. However, the mismatch is restricted to only one region. The authors should discuss why is the effect limited to only one region.

optix-Gal4 is expressed in the dorsal and ventral parts of the brain showing stronger signals in the dorsal part (Fig. 7e). When RNAi was induced under the control of *optix-Gal4*, the stronger phenotype was usually observed in the dorsal part of the brain (Line 365-367). We quantified the size of the overlap between Run- and Bsh-positive areas. The overlap was significantly increased in the dorsal RNAi area compared with the control areas (Fig. 7g-i).

These defects were restricted to the dorsal part of the brain, most likely due to the stronger expression of *optix-Gal4* in the dorsal brain (Fig. 7e).

4. Some parts of the discussion are unclear – in particular the paragraph starting at row 350 (the point is unclear),

Since it just summarized the published results, we shortened the paragraph (line 390-392).

According to the previous literatures, the upregulation of DI expression may induce the clustering of N and DI^{14, 34, 35}, which leads to an acute suppression of N signal activity via cis-inhibition (Fig. 6o).

the paragraph starting at row 394 (the point about the co-repressor seems irrelevant or at least unclear),

We removed this paragraph since we did not directly test the behaviors of Su(H) and H in this study.

and the last paragraph about NPC oscillation in vertebrates (is the point simply to say there are different mechanisms in different neuronal processes?)

Since the roles of cis-inhibition in mammalian NSC differentiation have not been studied so far, we changed this paragraph as follows (Line 432-438):

In the neural progenitor cells (NPCs) of the developing cerebral cortex, the temporal dynamics of N activity also plays important roles in the temporal patterning of neurogenesis and gliogenesis⁵³. In this process, the basic helix-loop-helix transcription factors show oscillatory expression in NPCs and N signaling appears to perform lateral inhibitory feedback during NPC differentiation. The roles of *cis*-inhibition in this process remain largely elusive. It will be interesting to see how the molecular

mechanisms revealed in the current study are conserved in a wide variety of developmental processes.

Minor issues:

5. The authors should also discuss whether Delta activity is inhibited by cis-interaction with Notch (e.g. Becam et al 2010).

The following sentences were added (line 422-424).

Although DI function is thought to be inhibited when DI and N interact in *cis*¹¹, we did not include this reaction in the model, because we do not have any observation that suggests *cis*-inhibition of DI in the course of the proneural wave progression.

6. OPC in row 409 is not defined

We do not use 'OPC' in the revised manuscript.

REVIEWERS' COMMENTS

Reviewer #1 (Remarks to the Author):

The authors have addressed my comments appropriately.

The paper remains quite difficult to read but it makes its points convincingly. It should be published
I only have one remaining minor point:

Minor point #7. The authors responded that "The neuronal type TmY14 has not been documented as far as we know based on its projection pattern in the medulla, lobula and lobula plate."
However, TmY14 was identified by Takemura et al, Nature 2013. Their connections in the medulla are available online.

A Tango-trace paper of Richard Axel (Neuron 2014) talks about TmY14 but do not call them like this. Cosmanescu et al., report about DIPs expression mentions medulla neurons amongst which TmY14. TmY14 anatomy has also been nicely described in Shinomiya et al., Frontiers in Neural Circuits 2019 "The Organization of the Second Optic Chiasm of the Drosophila Optic Lobe". I quote:

"The cell TmY14 (Takemura et al., 2013) is actually a visual projection neuron (VPN) morphologically distinct from other TmY cell types (Figure 3C). While the cell body resides in the medulla cortex, its cell body fiber penetrates the medulla without branching or making synapses, and bifurcates at the surface of the proximal medulla. One of the neurites further branches into multiple processes, which project to the medulla, lobula, and lobula plate neuropils. The other neurite projects to the central brain (A. Nern, unpublished observation). TmY14 cells have notably thicker axons (about 700–1500 nm in diameter) that are easily distinguishable from other fibers in the chiasm, and may correspond to the large-caliber axon profiles depicted by Braitenberg (1970). Unlike other TmY cells, each TmY14 cell may contribute up to three axons in the chiasm, at least one of which projects to both the lobula and lobula plate."

TmY14 is also described in Ozel et al, 2021, doi: 10.1038/s41586-020-2879-3.

Therefore, the authors must take this into consideration

Reviewer #2 (Remarks to the Author):

The authors have made all the modifications that I suggested. I have no further comments.

Reviewer #3 (Remarks to the Author):

I think that the authors sufficiently addressed the points that we raised. Although they did not provide direct evidence of how cis-inhibition by DI is mediated by intracellular trafficking using trafficking assays or EM, the genetics and imaging data added to this revised version much better support their mathematical model. I appreciated how they refined their immunostaining data and provided the requested quantifications. Moreover, the added sentences clarifying the explanation of the mathematical model will make the manuscript more readable for a broader audience.

Reviewer #4 (Remarks to the Author):

The authors addressed most of my comments and the comments of the other reviewers in a satisfactory manner. The additional explanation on the modeling and the expansion of the model to

include full length Notch are helpful. The new data supporting the trafficking model for cis-inhibition is useful and important. The discussion has improved considerably.

The overall findings are interesting and would contribute to the understanding of Neurogenesis and dynamics associated with cis-inhibition. Hence, I support acceptance of this manuscript.

Minor language editing are still needed.

Response to comments

Reviewer #1 (Remarks to the Author):

The authors have addressed my comments appropriately.

The paper remains quite difficult to read but it makes its points convincingly. It should be published

I only have one remaining minor point:

Minor point #7. The authors responded that "The neuronal type TmY14 has not been documented as far as we know based on its projection pattern in the medulla, lobula and lobula plate."

However, TmY14 was identified by Takemura et al, Nature 2013. Their connections in the medulla are available online.

A Tango-trace paper of Richard Axel (Neuron 2014) talks about TmY14 but do not call them like this. Cosmanescu et al., report about DIPs expression mentions medulla neurons amongst which TmY14.

TmY14 anatomy has also been nicely described in Shinomiya et al., Frontiers in Neural Circuits 2019 "The Organization of the Second Optic Chiasm of the Drosophila Optic Lobe". I quote:

"The cell TmY14 (Takemura et al., 2013) is actually a visual projection neuron (VPN) morphologically distinct from other TmY cell types (Figure 3C). While the cell body resides in the medulla cortex, its cell body fiber penetrates the medulla without branching or making synapses, and bifurcates at the surface of the proximal medulla. One of the neurites further branches into multiple processes, which project to the medulla, lobula, and lobula plate neuropils. The other neurite projects to the central brain (A. Nern, unpublished observation). TmY14 cells have notably thicker axons (about 700–1500 nm in diameter) that are easily distinguishable from other fibers in the chiasm, and may correspond to the large-caliber axon profiles depicted by Braitenberg (1970). Unlike other TmY cells, each TmY14 cell may contribute up to three axons in the chiasm, at least one of which projects to both the lobula and lobula plate."

TmY14 is also described in Ozel et al, 2021, doi: 10.1038/s41586-020-2879-3.

Therefore, the authors must take this into consideration

We would like to apologize that we carelessly used the name TmY14, which has been already used in some literatures. Actually, the neuron we identified in Fig. 7j is different from TmY14 published in the literatures. The biggest difference is that it does not

project to the central brain in contrast to TmY14. I talked with Dr. Shinomiya, who is an expert of this field and published the *Frontiers in Neural Circuits* 2019 paper, about this issue. He agreed that our neuron has not been published and is distinct from TmY14. However, he noted that it is identical to TmY16 neuron they are going to publish, according to its projection pattern. He agreed to describe the neuron as TmY16 and to cite his observation as a personal communication as quoted below.

>>Dear Makoto,

>>

>>I permit that you cite my unpublished work regarding the optic lobe neuron, the TmY16 cell, >>including its name and morphology, as a personal communication in your publication.

>>

>>Best wishes,

>>Kazunori

>>

>>--

>>Kazunori SHINOMIYA

>>FlyEM Project Team

>>HHMI Janelia Research Campus

>>19700 Helix Dr, Ashburn, VA 20147 USA

>>Phone: +1 (571) 209-4000 x3325

>><mailto:shinomiya@janelia.hhmi.org>

The text was modified accordingly as follows (Line 372-374).

The neuronal type TmY16 has not been documented based on its projection pattern in the medulla, lobula and lobula plate (Kazunori Shinomiya, personal communication).

Reviewer #2 (Remarks to the Author):

The authors have made all the modifications that I suggested. I have no further comments.

I would like to thank for the supportive comment.

Reviewer #3 (Remarks to the Author):

I think that the authors sufficiently addressed the points that we raised. Although they did not provide direct evidence of how cis-inhibition by DI is mediated by intracellular

trafficking using trafficking assays or EM, the genetics and imaging data added to this revised version much better support their mathematical model. I appreciated how they refined their immunostaining data and provided the requested quantifications. Moreover, the added sentences clarifying the explanation of the mathematical model will make the manuscript more readable for a broader audience.

I would like to thank for the supportive comments.

Reviewer #4 (Remarks to the Author):

The authors addressed most of my comments and the comments of the other reviewers in a satisfactory manner. The additional explanation on the modeling and the expansion of the model to include full length Notch are helpful. The new data supporting the trafficking model for cis-inhibition is useful and important. The discussion has improved considerably.

The overall findings are interesting and would contribute to the understanding of Neurogenesis and dynamics associated with cis-inhibition. Hence, I support acceptance of this manuscript.

Minor language editing are still needed.

I would like to thank for the supportive comments. Although there is a very tight word limit, we made some modifications in the text to make it easier to understand (Line 92, 109-111, 118, 219, 323).